# Psychiatric disorders comorbid with general medical illnesses and functional somatic disorders: The Lifelines cohort study

**Francis Creed***

Division of Psychology and Mental Health, University of Manchester, Manchester, United Kingdom

* francis.creed@manchester.ac.uk

## Abstract

### Objective

It is not clear why psychiatric disorders are more prevalent in the functional somatic syndromes than other general medical illnesses. This study assessed the correlates of psychiatric disorders in 3 functional syndromes and 3 general medical illnesses in a population-based sample.

### Methods

The Lifelines cohort study included 122,366 adults with relevant data for 6 self-reported conditions: irritable bowel syndrome (IBS), fibromyalgia, chronic fatigue syndrome (CFS), inflammatory bowel disease (IBD), rheumatoid arthritis (RA), and diabetes. For each condition the proportion with a DSM-IV psychiatric disorder was assessed. In a cross-sectional design, logistic regression identified at baseline the variables most closely associated with current psychiatric disorder in participants with a pre-existing medical or functional condition. In a separate analysis the prevalence of psychiatric disorder prior to onset of these conditions was assessed. This was a longitudinal study with psychiatric disorder assessed at baseline in participants who subsequently developed a general medical or functional condition between baseline and follow-up.

### Results

The prevalence of psychiatric disorder was higher (17–27%) in the functional somatic syndromes than the general medical illnesses (10.4–11.7%). The variables closely associated with psychiatric disorder were similar in the functional syndromes and general medical illnesses: stressful life events, chronic personal health difficulties, neuroticism, poor perception of general health, impairment of function due to physical illness and reported previous (lifetime) psychiatric disorder. The prevalence of psychiatric disorder prior to development of these disorder was similar to that of established disorders.

### Conclusion

Despite the difference in prevalence, the correlates of psychiatric disorders were similar in functional and general medical disorders and included predisposing and environmental

**Data Availability Statement:** "Data may be obtained from a third party and are not publicly available. Researchers can apply to use the Lifelines data used in this study. More information about how to request Lifelines data and the

conditions of use can be found on their website (https://www.lifelines.nl/researcher/how-to-apply)".

**Funding:** The author received no specific funding for this work.

**Competing interests:** The authors have declared that no competing interests exist.

factors. The increased rate of psychiatric disorder in functional somatic syndromes appears to be evident before onset of the syndrome.

## Introduction

Depression and anxiety are more prevalent in irritable bowel syndrome (IBS), fibromyalgia and chronic fatigue syndrome (CFS) than in comparable medical illnesses but the reasons for this are not known [1–8]. In chronic fatigue syndrome, for example, it has been suggested that depression may be secondary to the symptoms and impairment of CFS or that depression and CFS may share similar aetiological pathways with anxiety and depressive disorders [9]. The first suggestion reflects the fact that fibromyalgia may be associated with greater functional limitations than rheumatoid arthritis [10]. The second suggestion is supported by recent evidence showing that fibromyalgia, IBS and CFS carry a substantial genetic risk for anxiety and depressive disorders [11].

Prospective studies have shown a bidirectional relationship between psychiatric disorders and the functional somatic syndromes suggesting a particularly close relationship between these two types of disorder [12–16]. However, anxiety and depressive disorders are also frequent concomitants of general medical illnesses such as RA, Diabetes and IBD [17–21]. There is now some evidence that this relationship is also bidirectional with depression being a risk factor for these general medical illnesses as well as the other way round [22–29]. The relationship between psychiatric disorders and general medical disorders may be more similar to that between psychiatric disorders and functional somatic syndromes than previously considered. Clarifying this relationship requires more refined methodology than that used in previous studies.

In the existing literature the rate of psychiatric disorders among the physically ill tends to be overestimated because the studies have included clinical rather than population-based samples and have used self-administered questionnaires rather than research interviews to assess psychiatric disorders [2, 21, 23, 30, 31]. This prevents an accurate assessment of the relationship between psychiatric disorder and medical disorders [23, 32]. Furthermore, the effect of psychiatric disorder on treatment-seeking and on symptom reporting on questionnaires may be greater in the functional somatic symptoms than general medical disorders [7, 33]. The current study aimed to assess, in a general population-based sample, the prevalence of psychiatric disorders in 3 functional somatic syndromes and 3 general medical disorders using a research psychiatric interview. It also aimed to assess whether the correlates of psychiatric disorder were similar in functional somatic syndromes and general medical disorders. To my knowledge no previous study has performed such a large population study using research interviews to assess psychiatric disorder.

Prospective studies have shown that the prevalence of depression and other psychiatric disorders is higher than healthy controls both before and after the onset of inflammatory bowel disease, RA or diabetes and functional disorders [13, 17–27, 29, 30, 34–40]. The current study included a longitudinal analysis which assessed whether the prevalence of psychiatric disorder was similar before and after the onset of inflammatory bowel disease, RA, diabetes, IBS, CFS and fibromyalgia.

The first analyses in this study examined whether psychiatric disorder was closely associated with a) the symptoms and impairment which accompany a medical or functional disorder, and/or b) a predisposition to develop psychiatric disorder, including neuroticism, psychosocial stress and prior psychiatric disorder which would be evident before the onset of

the medical or functional disorder. Thus, for example, in rheumatoid arthritis it has been reported that the dominant predictors of self-reported depression are pain and fatigue and it is considered that these may be the main cause of the depression [41, 42]. Depression in fibromyalgia is also associated with severity of pain and degree of impairment but, in addition, depression in fibromyalgia is closely associated with psychological variables such as neuroticism, learned helplessness, low self-efficacy and with psychosocial problems [43–46]. These psychological variables are associated also with depression in the absence of physical illness [47–50]. No previous study has assessed, across a range of disorders, whether depression comorbid with physical illness is associated both with the symptoms and impairment of the physical illness and with the psychological variables of neuroticism, prior psychiatric disorder, low social support and stressful life events unrelated to physical illness. In this study it was hypothesised that the correlates of psychiatric disorders would be similar in participants with functional somatic syndromes and general medical illnesses.

Predisposition to develop psychiatric disorder might be genetic and/or relate to early development; in either case it would be evident in a high neuroticism score and previous episodes of psychiatric disorder [9, 11, 13, 16]. Environmental predisposing and precipitating factors of psychiatric disorder include few years of education, low income, less social support and experience of stressful life events and difficulties [48–50]. Poor perception of health, pain and impaired physical functioning would be more severe if the symptoms and impairment associated with the medical or functional disorder form a major factor leading to the psychiatric disorder.

The specific research question was: are the features associated with current psychiatric disorders in participants with IBS, CFS and fibromyalgia similar to, or different from, those associated with current psychiatric disorders comorbid with general medical illnesses? This question was addressed with both univariable and multivariate analyses. The psychiatric disorders were diagnosed by research interview but the general medical illnesses and functional disorders were self-reported.

It was not possible to perform a full prospective study as the Lifelines database did not include the MINI psychiatric interview at follow-up. It was possible, however, to identify new cases of the 6 disorders developing over the 2 ½ year follow-up period and identify the proportion of each who had a psychiatric disorder prior to the development of the medical or functional condition.

## Materials and methods

### Study design and participants

The data used in this study came from the Lifelines study, a multi-disciplinary prospective population-based cohort study examining in a unique three-generation design the health and health-related behaviours of 167,729 individuals living in the north of the Netherlands. The study assessed at baseline a broad range of biomedical, socio-demographic, behavioural, physical and psychological variables which may contribute to future health outcomes [51, 52]. People with low educational attainment and smokers are somewhat under-represented in the Lifelines cohort but otherwise it is representative of the total Dutch population [53]. Exclusion criteria were: severe psychiatric mental disorders (e.g. schizophrenia), severe physical illness, insufficient fluency in the Dutch language to complete questionnaires, inability to visit the general practitioner and limited life expectancy (<5 years). The participants were recruited between 2006 and 2013 and data collected at baseline were used for this study. Participants attended one of several study centres where questionnaires were completed on paper at the baseline assessment. A number of specific medical measurements (e.g. BMI, arthrometry)

were made; this was followed by the MINI interview (see below). Written informed consent was obtained from all participants. The study followed the guidelines of the Declaration of Helsinki and all procedures involving human subjects were approved by the UMCG Medical ethical committee under number 2007/152.

The present study included respondents who were 18 or more years of age at baseline, who had completed the questionnaire items relevant to IBS, CFS, fibromyalgia, inflammatory bowel disease, rheumatoid arthritis or diabetes and who had completed the detailed MINI neuropsychiatric interview (see below). The first set of analyses formed a cross-sectional study using baseline data only. The correlates of psychiatric disorder were determined among participants with pre-existing medical or functional conditions. The second, longitudinal study examined psychiatric disorder that was present at baseline before the onset of one of the medical or functional conditions; the 6 diagnostic groups were defined on the basis of medical or functional conditions which developed between baseline and first or second follow-up assessment.

**Measures.**  The baseline variables were chosen from the comprehensive Lifelines questionnaire if they were related to the known risk factors for depressive or anxiety disorders. These included socio-demographic variables, general medical illnesses, functional somatic syndromes, previous (lifetime) psychiatric disorders, stress, social support and impairment.

*Socio-demographic* features included sex, age, duration of formal education, marital and employment status and level of income.

*General medical illnesses*: Respondents were asked to indicate from a list of 30 current or past general medical illnesses which ones they have had and which medications they were taking at baseline. The questionnaire did not establish whether the diagnosis was made by a doctor. The total number of medical disorders was used as a measure of comorbidity. It included IBS, fibromyalgia and chronic fatigue syndrome. At follow-up participants were asked "Since the last assessment have you developed. . .(list of medical and functional conditions)". Participants were classified as a new onset of one of the 6 conditions if they had not recorded it at baseline and they recorded it at either follow-up.

*Functional somatic syndromes*: Participants were asked if they had had irritable bowel syndrome, chronic fatigue syndrome, fibromyalgia.

*Psychiatric disorders*: This was investigated in two ways. The questionnaire asked respondents to indicate which of the following psychiatric disorders they have had: depression, anxiety, burnout, panic disorder, social phobia, agoraphobia, obsessive compulsive and eating disorders. Such self-reported lifetime psychiatric disorders are reasonably accurate, especially for depressive disorders, but may be under-reported [54, 55]. In addition, the majority of Lifelines participants were interviewed by a trained medical professional using the MINI- International Neuropsychiatric Interview (MINI) 5.0.0 [56]. The MINI is a brief structured interview for diagnosing psychiatric disorders; in this study it was used to assess the presence of major depressive disorder (MDD), dysthymia, panic disorders, agoraphobia and general anxiety disorder (GAD) according to DSM-IV criteria [56]. Participants who had one or more of these diagnoses were categorised as having a psychiatric disorder and were compared with the remainder. Details of the use of the MINI have been reported previously [7, 30] and are available at https://www.lifelines.nl/researcher/data-and-biobank.

*Stress*: Recent stress was assessed using the List of Threatening Experiences (LTE) and Long-term Difficulties Inventory [57, 58]. A total score of the number of events and difficulties was calculated after excluding the items concerning stress caused by a recent chronic illness affecting the participant. The latter was quoted separately as a chronic personal illness difficulty so that it could be analysed separately from other forms of stress. The respondent was asked if their experience of their health was stressful (e.g. regularly ill, chronically ill) and they

responded on a three point scale; (0 = not stressful, 1 = slightly stressful, 2 = very stressful; the mean score was used in the analysis [58]. The total score for threatening life events and long term difficulties therefore included serious illness or death of a close relative and problems occurring in work, close relationships, financial, housing etc. A high score represented greater stress.

*RAND instrument* Current health status was assessed using the RAND 36-Item Health Survey General Health scale [59]. Four of the 8 scales were used as they were relevant to this study: Physical Function (10 questions) and Role Physical (4 questions) measured the degree to which daily life is restricted by physical limitations. Bodily pain (2 questions) assessed impairment due to pain. General Health (4 questions) assessed how ill the individual feels. Each question is scored on a 3, 4 or 5-point Likert scale. The scores are standardised and in the results presented here a low or negative score indicates greater impairment. Cronbach's alpha for these items in this sample were: Physical Functioning = 0.87, Role functioning—Physical = 0.89, Bodily Pain = 0.81, General Health = 0.80.

*Sleep*: The Pittsburgh Sleep Quality Index (PSQI) is a self-rated questionnaire which assesses sleep quality and disturbances over a 1-month time interval [60]. A high score indicates poor quality sleep.

*Neuroticism* This was assessed using the Revised NEO Personality Inventory (NEO PI-R) [51]. The Lifelines questionnaire included the facets of anger/hostility, self-consciousness, impulsivity, and vulnerability. Each facet is assessed with eight items, scored on a five-point Likert scale that ranges from strongly disagree to strongly agree [61].

*Social Appreciation* was assessed using the Social production function measure (SPF-IL). This is a 15 item scale asking about 6 dimensions of social support; each item is scored on a 4 point scale and a high score represents good social support [62].

## Outcome

For each of the six conditions the analyses compared the participants with and without a current psychiatric disorder and the variables associated with current psychiatric disorder were identified using univariate and multivariate analyses. The latter was repeated with the 3 medical disorders combined as one group and the 3 functional somatic syndromes as another group so the associated features of functional somatic syndromes and three general medical illnesses could be compared.

## Statistical analyses

All analyses were performed on SPSS statistics 25. The number of participants with each of the six conditions who had a psychiatric disorder was ascertained.

**Univariable analysis.** The first set of analyses compared participants with and without psychiatric disorder for each of the six conditions. The significance of differences was assessed using chi square or ANOVA as appropriate. The value of p was set at 0.001 in view of the large number of variables that were assesses as correlates of psychiatric disorder.

## Multivariate analysis

Logistic regression analysis with backward elimination of variables was used to determine the variables associated with psychiatric disorder. This analysis was performed for each of the six conditions separately. The dependent variable was psychiatric disorder, present or absent. The independent variables were all the variables listed in Table 1 except the 5 DSM-IV diagnoses and 3 general medical diagnoses (inflammatory bowel disease, rheumatoid arthritis and diabetes, which were included in the number of general medical illnesses score). The participants

**Table 1. Characteristics of the sample.**

|  | N = 122,366 |
|---|---|
| **Categorical variables** |  |
| Female sex | 58.5% |
| Few years of education (secondary or less) | 28.8% |
| Married or cohabiting | 81.3% |
| In full time employment | 40.8% |
| Unable to work through illness | 2.8% |
| Low income (lowest quintile) | 14.9% |
| Smoking past or present | 20.9% |
| IBS | 9.5% |
| CFS | 1.1% |
| Fibromyalgia | 3.1% |
| Inflammatory bowel disease | 0.9% |
| Rheumatoid arthritis | 2.0% |
| Diabetes | 2.3% |
| Life psychiatric disorder (self-report) | 17.9% |
| *DSM-IV diagnoses (MINI)*: |  |
| Generalised anxiety disorder | 4.3% |
| Panic disorder | 3.1% |
| Dysthymia | 1.2% |
| Major depressive disorder | 1.9% |
| Agoraphobia | 3.2% |
| **Continuous variables Mean (sd)** |  |
| Age | 44.2 yrs (12.6) |
| Life events and diffs score | 2.6 (1.7) |
| No. of general medical illnesses | 1.5 (0.8) |
| Chronic illness difficulties | 1.25 (0.5) |
| Neuroticism | -4.9 (1.7) |
| Social appreciation score | 25.0 (3.5) |
| PSQI score | 3.9 (2.2) |
| *RAND items*: |  |
| General health | 68.6 (12.3) |
| Bodily pain | 85.3 (18.3) |
| Physical functioning | 91.1 (13.4) |
| Role physical | 87.3 (26.1) |

PSQI = Pittsburgh Sleep Quality Inventory

with IBS, fibromyalgia or CFS were combined to form a single "functional somatic syndrome" group. Those with inflammatory bowel disease, diabetes, and rheumatoid arthritis were combined into a single "general medical illnesses" group, and the logistic regression analysis was repeated to identify the variables associated with psychiatric disorder in these two large groups.

# Results

## Sample

Of the Lifelines cohort, 152,180 participants were 18 + years of age (up to 93 years), and 122,366 of these reported the relevant data concerning medical conditions and functional

**Table 2. Number (%) of participants with one or more DSM-IV psychiatric disorders for six conditions.**

| Diagnosis | Participants with a medical or functional diagnosis | Participants without a general or functional diagnosis | |
|---|---|---|---|
| Diabetes | 10.9% | 8.9% | <0.001 |
| | 302/2761 | 10,607/119,360 | |
| RA | 11.7% | 8.9% | <0.001 |
| | 287/2452 | 10,654/119,914 | |
| Inflammatory Bowel Disease | 10.4% | 8.9% | ns |
| | 114/1096 | 10,827/121,270 | |
| IBS | 16.7% | 8.1% | <0.001 |
| | 1921/11,489 | 9020/110,877 | |
| CFS | 27% | 8.7% | <0.001 |
| | 379/1406 | 10,562/120,960 | |
| Fibromyalgia | 19.4% | 8.6% | <0.001 |
| | 724 /3733 | 10,217/118,633 | |

syndromes and had been interviewed using the MINI interview. The characteristics of the sample are shown in Table 1. Of the six conditions, IBS was the most common (n = 11,489) and inflammatory bowel disease the least (n = 1096).

The proportion of each condition with a psychiatric disorder is shown on Table 2. The proportion was highest in CFS (27%), intermediate in IBS and fibromyalgia and lowest in inflammatory bowel disease, RA and diabetes (10.4–11.7%). For each condition, except inflammatory bowel disease, the prevalence of psychiatric disorder was greater in those with a medical or functional disorder compared to those without (Table 2). Fig 1 shows the DSM-IV psychiatric disorders for each of the 6 conditions; data are shown for females only as there was a significantly higher rate of psychiatric disorder in females and the proportion of females varied in each diagnostic group (fibromyalgia 91.5% were females, IBS 81%, CFS 70.3%, IBD 64.5%, RA 63.5%, Diabetes 53.7%). The pattern of psychiatric diagnoses was similar in all conditions with generalised anxiety disorder being the most common and dysthymia the least common (Fig 1). The prevalence was lower in men but the pattern was similar (S1 Table).

## Univariable analysis of variables associated with psychiatric disorder

Table 3 shows for inflammatory bowel disease the differences between the participants with and without a psychiatric disorder. Comparable data for the other 5 conditions are shown in the (S2–S6 Tables). Overall, the univariable analyses showed that psychiatric disorder is associated with nearly all the variables in these tables.

Figs 2–6 show these data in diagrammatic form. Psychiatric disorder was associated with female sex in IBS, inflammatory bowel disease, rheumatoid arthritis and diabetes but not fibromyalgia or CFS (Fig 2). Nearly all (91.5%) participants with fibromyalgia were female but the corresponding proportion of participants with CFS was 70.5%.

Neuroticism score was higher in participants with current psychiatric disorder in all 6 conditions but it was especially elevated in CFS (Fig 3). Chronic health difficulties affecting the participant were significantly associated with psychiatric disorder in all 6 conditions (Fig 4). The life events and difficulties score (excluding personal illness) was associated with comorbid psychiatric disorder but the score was high in CFS even in the absence of psychiatric disorder (Fig 5). The proportion with a lifetime history of psychiatric disorder was greater in the participants with a functional somatic syndrome (65.3–76%) compared to the general medical illnesses (50.5–54%) with the proportion being greatest in CFS (Fig 6).

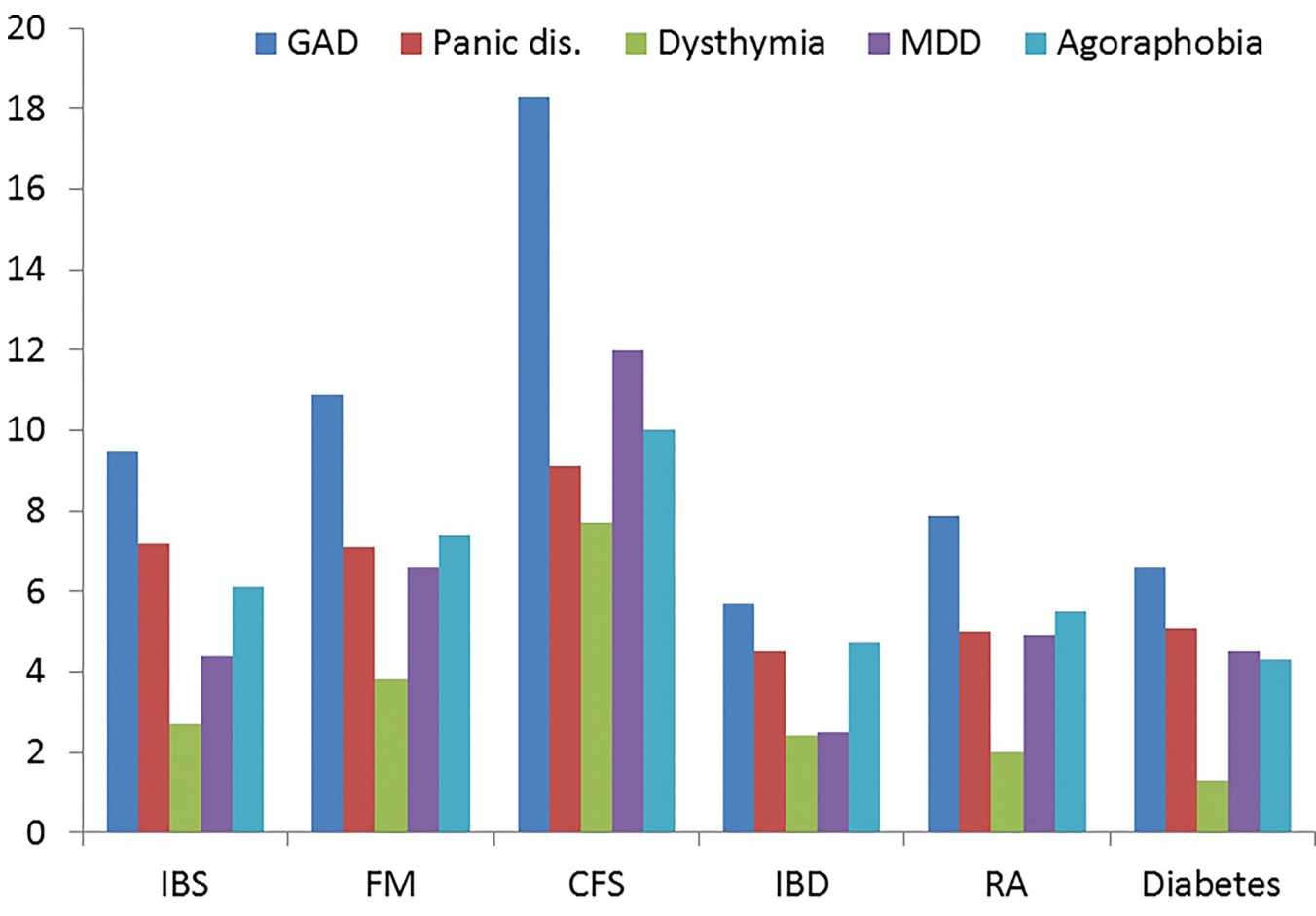

(GAD= generalised anxiety disorder, MDD= major depressive disorder)

**Fig 1. Prevalence of the five psychiatric disorders in women in the six conditions.**

### Logistic regression analyses to adjust for all potential correlates

The results of the logistic regression analyses for each condition are shown in Table 4, which shows Odds Ratios (OR) and 95% confidence intervals. It can be seen that neuroticism is a predictor of psychiatric disorder in all conditions except diabetes. Life events and difficulties score was a predictor of psychiatric disorder in all conditions except inflammatory bowel disease and poor perception of general health in all conditions except rheumatoid arthritis. Lifetime psychiatric disorder was a predictor in all conditions.

The predictors of psychiatric disorder were similar in the two logistic regression analyses including a) functional somatic syndromes and b) general medical illnesses (Table 5). In both the predictors were: stressful life events and difficulties, chronic personal health difficulties, neuroticism, poor perception of general health, impairment of function due to physical illness and reported previous psychiatric disorder (Table 5).

### Prevalence of psychiatric disorders in participants who later developed one of the 6 conditions

The prevalence of psychiatric disorder at baseline in participants who reported development of one of the conditions during the follow-up period, having been free of it at baseline, was as

Table 3. Participants with inflammatory bowel disease.

| | No psychiatric disorder | Psychiatric disorder | P value |
|---|---|---|---|
| | N = 982 | N = 114 | |
| **Categorical variables** | | | |
| % female | 62.8% | 78.9% | 0.001 |
| Few years education | 30.2% | 28.9% | ns |
| Work f/t | 34.7% | 24.6% | 0.030 |
| Off sick | 7.8% | 19.3% | <0.001 |
| Low income | 13.9% | 23.7% | 0.006 |
| IBS | 11.7% | 24.6% | <0.001 |
| Fibromyalgia | 4.1 | 13.2 | <0.001 |
| Lifetime psychiatric disorder | 14.9% | 52.6% | <0.001 |
| **Continuous variables mean (sd)** | | | |
| Age | 46.4 (11.3) | 44.0 (10.3) | 0.033 |
| Life events and difficulties score | 2.6 (1.7) | 3.4 (1.7) | <0.001 |
| No. of general medical illnesses | 1.7 (0.9) | 1.9 (0.9) | ns |
| Chronic illness difficulties | 1.6 (0.6) | 2.0 (0.7) | <0.001 |
| Neuroticism | -4.9 (1.8) | -4.2 (1.9) | <0.001 |
| Social appreciation score | 24.9 (3.5) | 23.4 (3.9) | <0.001 |
| PSQI score | 4.0 (2.3) | 5.2 (3.1) | <0.001 |
| RAND items: | | | |
| General health | 62.6 (14.7) | 53.8 (16.2) | <0.001 |
| Bodily pain | 78.3 (21.3) | 67.2 (26.7) | <0.001 |
| Physical functioning | 81.3 (18.6) | 63.8 (21.7) | <0.001 |
| Role physical | 78.8 (29.3) | 54.8 (43.5) | <0.001 |

PSQI = Pittsburgh Sleep Quality Inve29.3ntory

follows: IBS 14.7% (205/1390), inflammatory bowel disease 16.5% (27/164), fibromyalgia 18.7% (141/754), rheumatoid arthritis 11.6% (91/782), chronic fatigue syndrome 26.9% (90/334) and diabetes 8.5% (68/801). These proportions are very similar to those recorded in participants with established conditions (Table 2) who formed the main part of this study, except inflammatory bowel disease which was the least common condition.

## Discussion

This study found that, in both univariate and multivariate analyses, the correlates of psychiatric disorders were similar in participants with functional somatic syndromes and general medical illnesses. The study found that in both general medical disorders and functional syndromes psychiatric disorder was correlated with impairment and bodily pain as well as with predisposing and precipitating factors for psychiatric disorder such as past psychiatric history, neuroticism and poor social support and stressful life events. The study confirmed that the prevalence of psychiatric disorder in RA, IBD and diabetes (10.4–11.7%) was lower than that in IBS, CFS and fibromyalgia (16.7–27.9%) but higher than that in participants without these illnesses (8.1–8.9%). The higher prevalence of psychiatric disorder in functional somatic syndromes than general medical illnesses appears to be related primarily to the higher rate of prior (lifetime) psychiatric disorders and, possibly, in higher neuroticism score in CFS. The longitudinal analysis suggested that the prevalence of psychiatric disorders prior to the onset of IBS, CFS and fibromyalgia (14.7–26.9%) was also higher than that preceding RA, IBD

% female

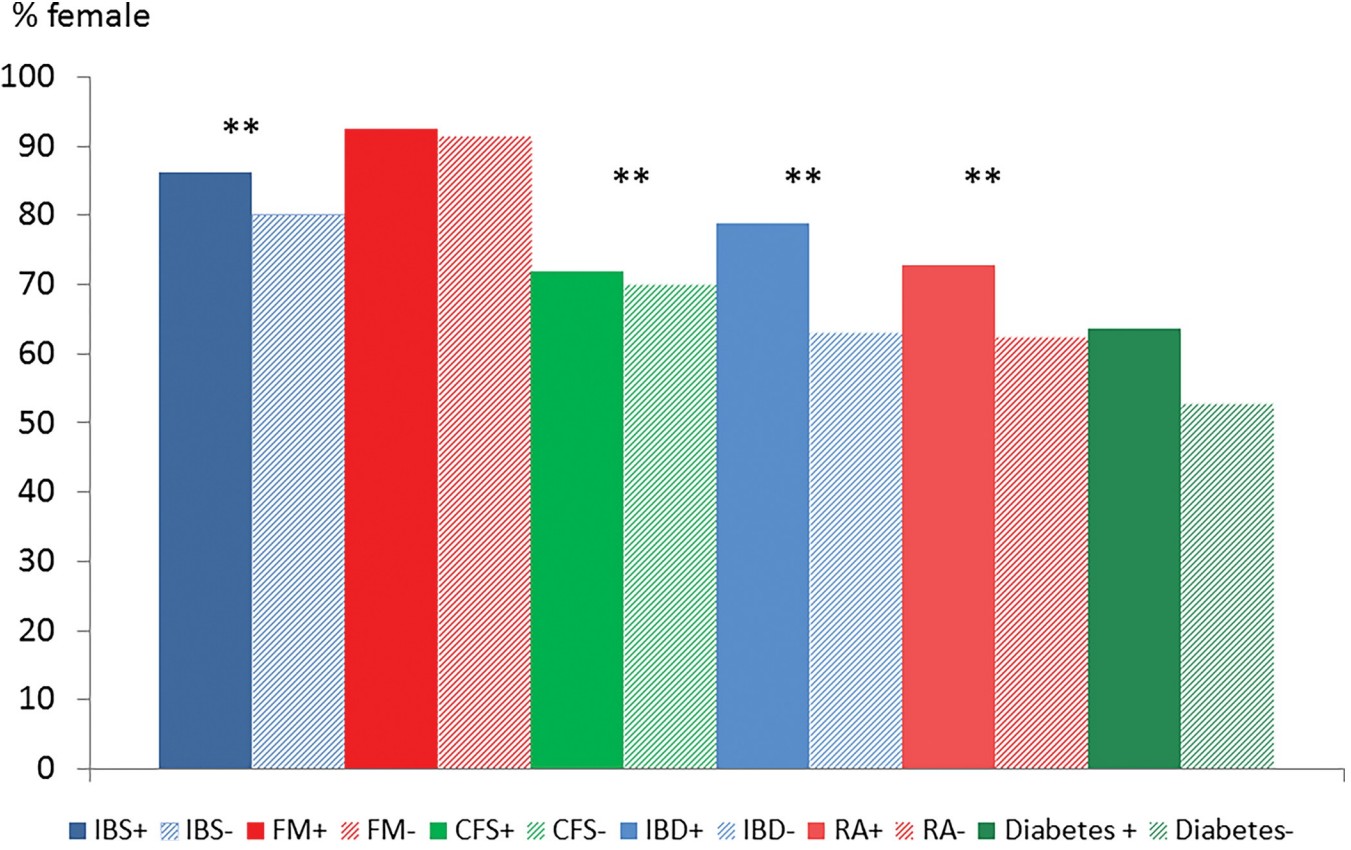

**Fig 2. Proportion of females in participants with (+) and without (-) psychiatric disorder by condition.**

and diabetes (8.5%-16.5%) and suggests that psychiatric disorder generally precedes, rather than follows the onset of all 6 conditions.

## Comparison with previous literature

A systematic review of depression and anxiety comorbid with inflammatory bowel disease reported similar rates of psychiatric disorder to those found in this study with the majority developing prior to onset of IBD [30]. Other studies confirmed that female sex and aggressive or more active disease were independently associated with depression in inflammatory bowel disease as was concurrent IBS [63–65]. Systematic reviews have found the prevalence of inter-view-rated depressive disorder in rheumatoid arthritis to be slightly higher than this study; it was associated with medical comorbidities such as cancer, chronic kidney disease and stroke but previous studies have not included the broad range of other correlates included in the current study [6, 19, 42, 66].

One review found a rate of psychiatric disorder in type 2 diabetes of 7% in studies with a low risk of bias and the correlates were female sex, lower age, few years of education, complications of diabetes, living alone, use of insulin [20]. Another systematic review of depression in diabetes cautioned against drawing conclusions from existing studies because of variable quality; in particular most were not population-based sample and most failed to control adequately for confounders [23]. One interesting study found that the prevalence of depression was raised in diabetes even if the diabetes was previously undiagnosed; anxiety was more prevalent in diagnosed diabetes possibly as a reaction to the knowledge and management of the illness [67].

Neuroticism score

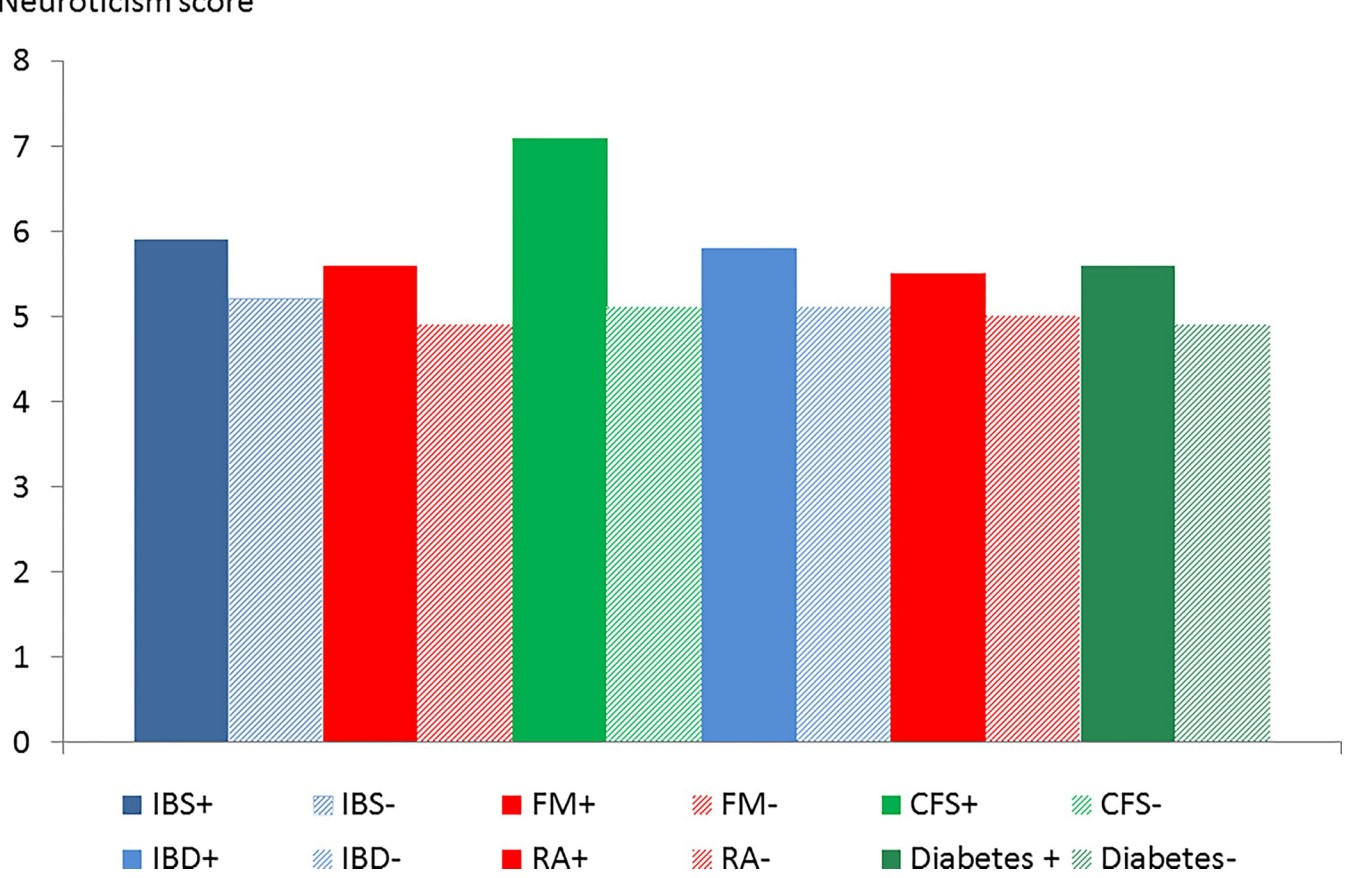

**Fig 3. Neuroticism score in participants with (+) and without (-) psychiatric disorder by condition.**

Like this study a previous review found the highest rate of depressive and anxiety disorders in CFS, the lowest in fibromyalgia, with IBS in an intermediate position [1]. Two population-based studies, one using the same database as the present study, have reported a higher prevalence of psychiatric disorder in participants with IBS, fibromyalgia or CFS than healthy comparison participants but neither study included data concerning other medical disorders [7, 8]. Both studies found the rate was higher was higher in CFS than IBS and fibromyalgia.

The most relevant systematic review of fibromyalgia found results very similar to the present study: the prevalence of interview-assessed current major depressive disorder was 25% whereas life-time prevalence was 65% [2]. A large study of CFS using a structured interview found 46% had a depressive or anxiety disorder, or both; this study also reported that comorbid depression was associated with a greater number of medical comorbidities [4].

A large population-based study, comparable with the present one, found that participants with self-reported fibromyalgia were twice as likely as healthy controls to have a recent or life-time anxiety / depressive disorder; self-reported fibromyalgia was also positively associated with many comorbid medical conditions [32].

## Temporal relationship of psychiatric disorders and medical or functional disorders

It was mentioned in the introduction that depression or other psychiatric disorders may precede the onset of inflammatory bowel disease, RA or diabetes [22–24, 26, 27, 29, 34–38]. It

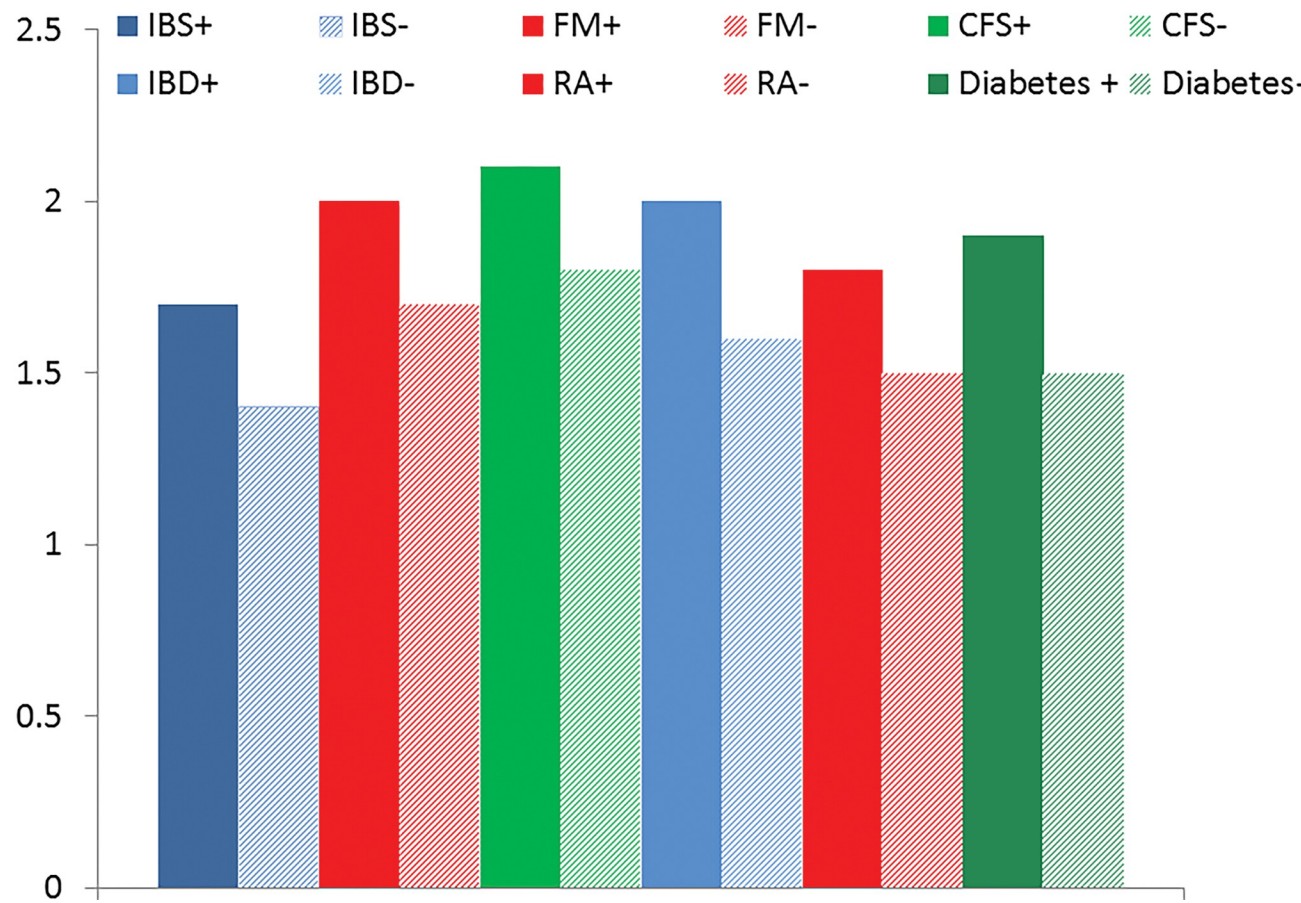

**Fig 4. Chronic personal health difficulty score in participants with (+) and without (-) psychiatric disorder by condition.**

may also develop after the onset of all these conditions [20, 22, 24–27, 30, 39, 40]. This bi-directional relationship has been shown also in the functional somatic syndromes and this has led some authors to imply that the relationship is causal in relation to IBS. Thus when depression precedes IBS onset the disorder was described as "brain-gut" IBS and when the depression followed IBS onset it was described as "gut-brain" IBS, implying that the IBS causes the depression and vice versa [13, 39].

Few studies have addressed the time course of psychiatric disorder preceding onset of these disorders. One large, population-based study found that the consultation rate for psychological disorders rose sharply for 2 years before the diagnosis of IBS, CFS or Fibromyalgia suggesting anxiety and depression preceded the onset of these syndromes but the rise also continued for 2 years after the onset [34]. The latter could indicate psychiatric disorder developing after the diagnosis but it could reflect referral for psychological treatment of the functional somatic syndromes as the study was concerned only with healthcare seekers [34]. In Rheumatoid arthritis a high incidence rate was observed in people with depression and the rate was highest when depression had been present for at least 3 years [28]. The onset of inflammatory bowel disease has been associated with depression which had been present for 5 years or more [38]. These findings suggest that it may be chronic depressive disorder which is associated with the

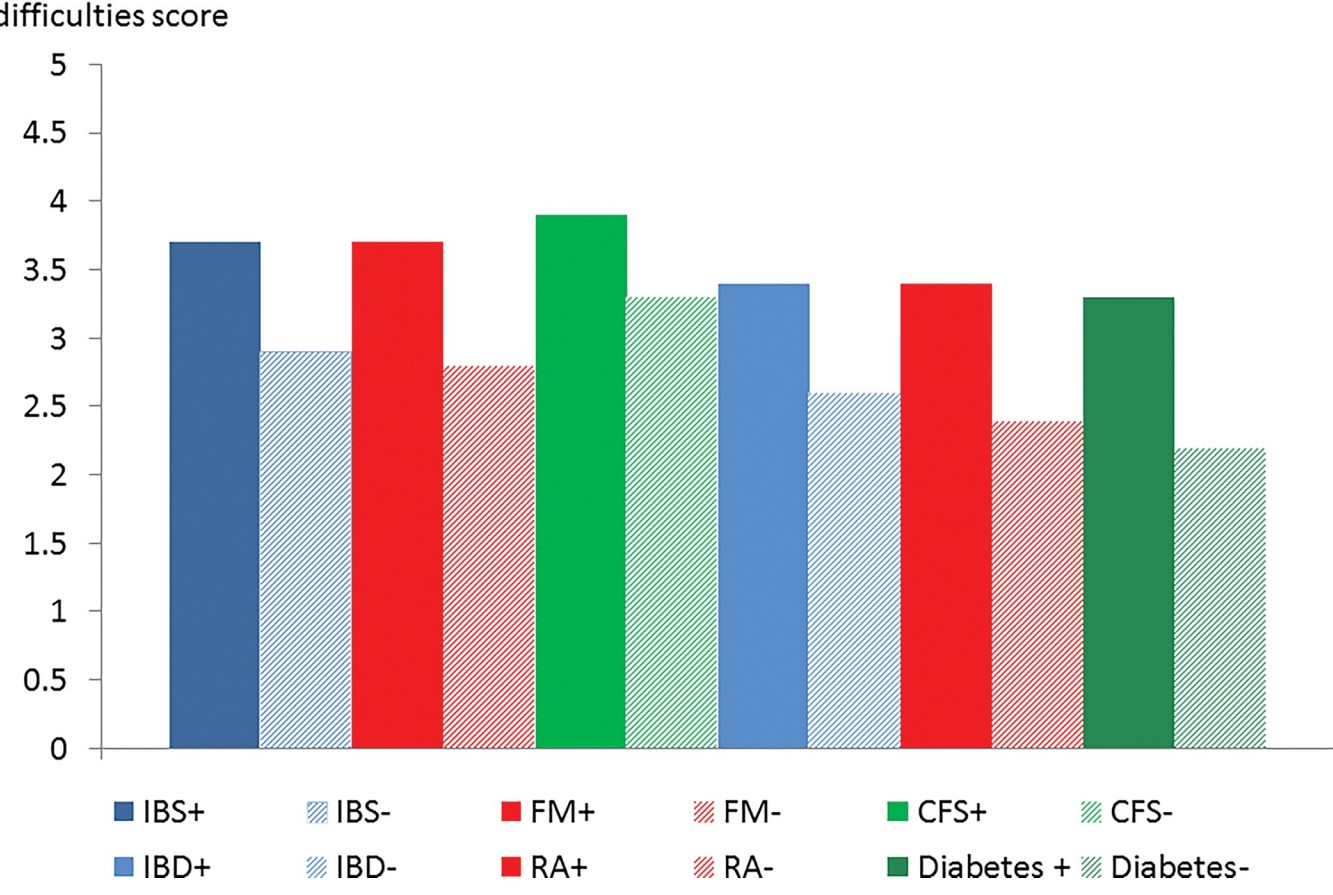

**Fig 5. Life events and difficulty score in participants with(+) and without (-) psychiatric disorder by condition.**

subsequent onset of these medical disorders. Similarly IBS onset has been associated with anxiety and depression that was diagnosed up to 6 years previously [68]. Depression onset is also noted to be particularly high in the first year after IBS onset [40]. These protracted time periods need to be taken into consideration when considering possible causal links.

## Multifactorial view of anxiety / depression and medical or functional disorders

Two studies help us to put into perspective these findings regarding psychiatric disorders and disease onset. Donnachie showed that the marked increase in consultation rate prior to the first diagnosis of IBS, fibromyalgia and CFS occurred for psychological disorder but was accompanied also by increases in consultation for somatic symptoms and for infectious gastroenteritis [34]. The strength of these associations varied somewhat between the three functional somatic syndromes; for infectious gastroenteritis, the odds ratios were significantly greater for IBS than CFS or fibromyalgia, whereas for psychological disorders they were greater for CFS than Fibromyalgia and IBS [34]. In other words, depression must be considered alongside other risk factors if we are to understand their relationship to disease causation.

In a detailed prospective study of diabetes individuals exposed to both behavioural and metabolic factors were at highest risk of diabetes onset; depression and anxiety contributed further

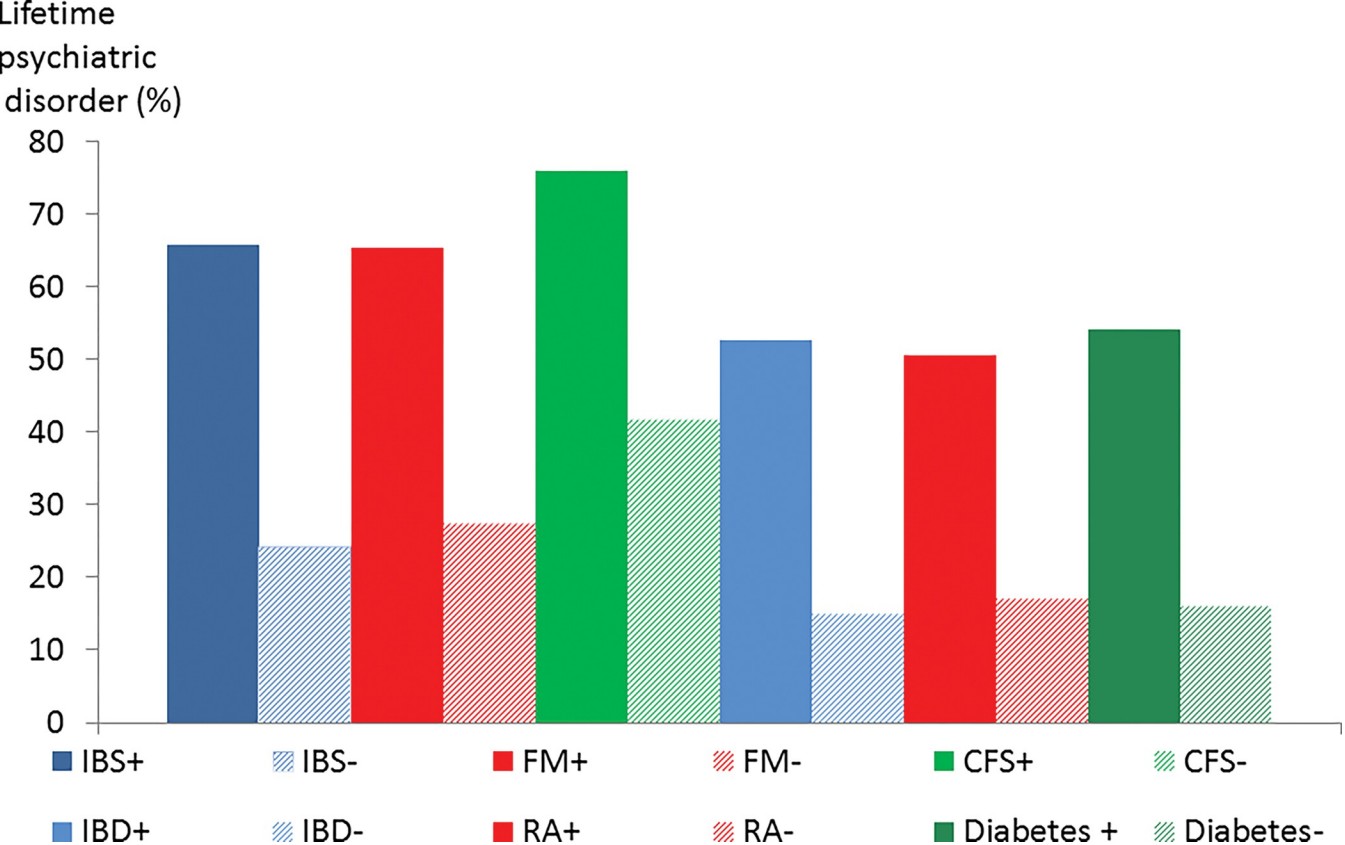

**Fig 6. Lifetime psychiatric disorder in participants with(+) and without (-) psychiatric disorder by condition.**

to population attributable fractions, but they did so only in conjunction with the metabolic and behavioural risk factors [69]. No diabetes cases in the overall sample were attributable to the presence of psychological risk factors alone [69]. Thus further research should examine interactions between anxiety/depressive disorders and other aetiological factors.

Although psychiatric disorders have been shown to be associated with the onset of IBS, fibromyalgia and CFS, a recent study using the wide range of covariates in the Lifelines database found that psychiatric disorder was a predictor only of IBS [70]. This may reflect the existence of a subgroup of IBS onsets which are closely associated with psychiatric disorder [71]. Interactions between psychosocial variables and other aetiological factors need to be considered in future research of the functional somatic syndromes [72].

## Strengths and limitations

The strengths of this study include the large population-based sample, the wide range of covariates and the use of a research interview to assess psychiatric disorders. The functional syndromes and general medical illnesses were assessed independent of medical help-seeking, a limitation of the many population-based studies which rely on case register data; psychiatric disorders are one of the determinants of healthcare use.

The most important limitations are the self-reported nature of the functional syndromes and medical illnesses and the cross-sectional design, which prevents conclusions about the causation or temporality of the association between psychiatric disorder and these disorders. A prospective study is needed which assesses, by research interview, the psychiatric disorders

**Table 4. Logistic regression analysis to demonstrate variables associated with presence of psychiatric disorder in each of the six conditions (blank means that the odds ratio was not significant).**

|  | IBS | IBD | Fibromyalgia | RA | CFS | Diabetes |
|---|---|---|---|---|---|---|
| n | 1623/9602 | 96/896 | 597/3027 | 217/1796 | 316/ 1166 | 221/1826 |
| Sex | 1.28 (1.07–1.53) | 2.06 (1.17–3.64) | ns |  |  |  |
| Age |  |  |  |  | 1.02 (1.003–1.03) |  |
| Low income |  |  | 1.42 (1.13–1.79) | 1.73 (1.19–2.510 |  |  |
| Work f/t |  |  |  |  | 0.70 (0.50–0.98) |  |
| Off sick | 1.04 (1.01–1.08) |  |  |  |  |  |
| Neuroticism | 1.14 (1.10–1.18) | 1.22 (1.06–1.41) | 1.10 (1.04–1.17) | 1.11 (1.01–1.22) | 1.21 (1.11–1.31) |  |
| Social appreciation score | 0.97 (0.95–0.99) |  | 0.93 (0.90–0.95) | 0.94 (0.91–0.98) |  |  |
| No. of general medical disorders |  |  |  | 1.23 (1.05–1.45) |  |  |
| Chronic fatigue syndrome. |  |  |  | 2.56 (1.34–3.87) |  | 2.16 (1.04–4.50) |
| Stressful chronic illness | 1.16 (1.04–1.29) |  | 1.22 (1.03–1.44) |  |  | 1.36 (1.04–1.78) |
| Life events and difficulties score | 1.19 (1.14–1.24) |  | 1.17 (1.09–1.25) | 1.20 (1.09–1.32) | 1.18 (1.06–1.30) | 1.19 (1.08–1.32) |
| General health | 0.85 (0.77–0.93) | 0.63 (0.43–0.92) | 0.69 (0.48–1.0) |  | 0.78 (0.62–0.97) | 0.77 (0.60–0.98) |
| Role Physical | 0.91 (0.85–0.97) | 0.78 (0.61–0.98) | 0.88 (0.80–0.97) | 0.82 (0.72–0.95) |  |  |
| Bodily pain |  |  |  |  |  | 0.83 (0.70–0.93) |
| Lifetime psychiatric disorder | 4.44 (3.94–5.01) | 5.73 (3.51–9.34) | 3.53 (2.88–4.32) | 3.45 (2.52–4.74) | 3.34 (2.46–4.53) | 3.89 (2.82–5.36) |

both before and after development of functional somatic syndromes and general medical illnesses. Such a study is unlikely in the near future as interview-determined psychiatric disorders are rare in large epidemiological studies; this makes the results of this study important. The preliminary data presented here indicate a rather similar prevalence of psychiatric disorder before and after the onset of the functional syndromes and general medical illnesses but

**Table 5. Logistic regression analysis to determine variables associated with presence of psychiatric disorder.**

|  | Functional Somatic Syndromes n = 12,362 of whom 2153 (17.4%) had psychiatric disorder | General medical illnesses n = 4376 of whom 513 (11.7%) had psychiatric disorder |
|---|---|---|
| Sex | 1.19 (1.02–1.38) | ns |
| Work f/t. | 0.85 (0.74–0.97) | ns |
| Low income | ns | 1.34 (1.05–1.72) |
| Neuroticism | 1.13 (1.10–1.17) | 1.11 (1.04–1.18) |
| Social appreciation score | 0.96 (0.94–0.97) | 0.97 (0.94–0.99) |
| No. of General medical illnesses | ns | ns |
| Chronic fatigue syndrome. | ns | 1.93 (1.22–3.06) |
| Chronic personal health difficulty. | 1.20 (1.10–1.31) | 1.16 (0.97–1.38) |
| Life events and difficulties score | 1.19 (1.15–1.23) | 1.18 (1.10–1.25) |
| RAND item General Health perception | 0.87 (0.79–0.94) | 0.76 (0.65–0.89) |
| RAND item Role Physical | 0.91 (0.86–0.96) | 0.86 (0.77–0.95) |
| Lifetime Psychiatric Disorder | 4.14 (3.70–4.60) | 4.07 (3.32–5.00) |

this is not the same as a prospective study of a single cohort. The study did not include other predisposing factors for psychiatric disorders, such as family history of psychiatric disorder and childhood abuse as these were not included in the Lifelines database. It is possible that the psychiatric disorder may have influenced some of the self-reported variables but such an influence should affect the functional syndromes and general medical illnesses equally.

## Conclusions and recommendations

Although this study confirmed a higher prevalence of psychiatric disorders in the functional somatic syndromes compared to 3 general medical illnesses it did not find any clear difference in the correlates of psychiatric disorders that might explain this difference. The pattern of correlations was similar for the variables indicating predisposition to develop psychiatric disorder and those reflecting pain and impairment of function. This would not be expected if there was a fundamental difference in the way that the psychiatric disorder developed, e.g. shared aetiology versus reaction to the symptoms and impairments.

The only clear difference was in the rate of prior (lifetime) psychiatric disorders which was greater in the functional somatic syndromes. The possible reasons for this lies outside the range of variables included in this study; genetic factors and childhood abuse might be two possible candidates [11, 73–76].

There are two recommendations arising as a result of this study The first is that future work concerning the relationship between psychiatric disorder and functional somatic syndromes must be considered alongside the same relationship with general medical disorders, in which depression may accompany the disorder without necessarily being a causal factor. This should help avoid premature conclusions that psychiatric disorder is a principal factor causing the functional somatic syndromes [70]. Secondly, in both research and clinical work we should be aware of possible interactions between psychiatric disorders and other behavioural and metabolic variables to identify the true role of anxiety and depression in the causation of the functional somatic syndromes.

## Supporting information

**S1 Table. Proportion with psychiatric disorder in the 6 diagnostic groups by sex.**
(DOCX)

**S2 Table. Participants with irritable bowel syndrome.**
(DOCX)

**S3 Table. Participants with rheumatoid arthritis.**
(DOCX)

**S4 Table. Participants with fibromyalgia.**
(DOCX)

**S5 Table. Participants with chronic fatigue syndrome.**
(DOCX)

**S6 Table. Participants with diabetes.**
(DOCX)

## Acknowledgments

This study was performed using data from the Lifelines project, University of Groningen Netherlands. It was only possible with the help and co-operation of the Lifelines management staff

and Judith Rosmalen, Rei Monden and Klaas Wardenaar of University Medical Center Groningen, who worked on a previous paper.

The Lifelines Biobank initiative has been made possible by subsidy from the Dutch Ministry of Health, Welfare and Sport, the Dutch Ministry of Economic Affairs, the University Medical Center Groningen (UMCG the Netherlands), University of Groningen and the Northern Provinces of the Netherlands. The author wishes to acknowledge the services of the Lifelines Cohort study and all the study participants.

**Ethical approval**. Approved by the Medical Ethical Committee of the University Medical Center Groningen.

## Author Contributions

**Conceptualization:** Francis Creed.

**Formal analysis:** Francis Creed.

**Methodology:** Francis Creed.

**Writing – original draft:** Francis Creed.

**Writing – review & editing:** Francis Creed.

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
