## [Decision Letter · Decision Letter 0]

7 Feb 2023

PONE-D-22-33955Psychiatric disorders comorbid with general medical illnesses and functional somatic disorders: The Lifelines cohort studyPLOS ONE

Dear Dr. Creed,

Thank you for submitting your manuscript to PLOS ONE. After careful consideration, we feel that it has merit but does not fully meet PLOS ONE’s publication criteria as it currently stands. Therefore, we invite you to submit a revised version of the manuscript that addresses the points raised during the review process.

We look forward to receiving your revised manuscript.

Kind regards,

Wudneh Simegn Belay, MSc

Academic Editor

PLOS ONE

Journal Requirements:

3. PLOS requires an ORCID iD for the corresponding author in Editorial Manager on papers submitted after December 6th, 2016. Please ensure that you have an ORCID iD and that it is validated in Editorial Manager. To do this, go to ‘Update my Information’ (in the upper left-hand corner of the main menu), and click on the Fetch/Validate link next to the ORCID field. This will take you to the ORCID site and allow you to create a new iD or authenticate a pre-existing iD in Editorial Manager. Please see the following video for instructions on linking an ORCID iD to your Editorial Manager account: https://www.youtube.com/watch?v=_xcclfuvtxQ.

Reviewers' comments:

Reviewer's Responses to Questions

**Comments to the Author**

1. Is the manuscript technically sound, and do the data support the conclusions?

Reviewer #1: Yes

Reviewer #2: Partly

2. Has the statistical analysis been performed appropriately and rigorously? 

Reviewer #1: Yes

Reviewer #2: Yes

3. Have the authors made all data underlying the findings in their manuscript fully available?

Reviewer #1: No

Reviewer #2: Yes

4. Is the manuscript presented in an intelligible fashion and written in standard English?

Reviewer #1: Yes

Reviewer #2: No

5. Review Comments to the Author

Reviewer #1: This study investigates determinants of psychiatric disorders in patients with functional vs. general medical illnesses. The author uses data from the Lifelines cohort, a large population-based study. The sample includes the impressive number of 122.366 adults. The results show that the variables associated with psychiatric morbidity were quite similar for patients with functional and general medical illness. A longitudinal corollary analysis showed that the prevalence of psychiatric disorder prior to the development of the functional and general medical conditions were quite similar to that of the participants with already established conditions. The author concludes that the increased rate of psychiatric disorder in patients with functional somatic syndromes seems to be evident before onset of the syndrome, which might be due to genetic as well as environmental factors.

This is is an interesting study with some unique aspects. It is a very large population-based sample. The study assessed functional syndromes as well as general medical conditions. Psychiatric diagnoses were established using a structured psychiatric interview with the majority of the participants. The main limitation refers to the self-report of FSS and general medical conditions. Furthermore, only a subset of mental disorders were assessed during the psychiatric interview.

Despite these limitations, this is a very important study, as it provides direct comparison between people with medically unexplained and medically explained conditions from the general population. This allows a more unbiased evaluation of possible determinants of psychiatric comorbidity in these two groups of disorders. The DSM-5 has abandoned the distinction between medically explained and medically unexplained symptoms. However, the phenomenen of medically unexplained symptoms won't vanish only because of the new diagnostic category in the DSM-5. Patients and clinicans will continue to differentiate between these two phenomenon in clinical practice. Thus, results on differences and similarities in these two groups are of high interest.

This is a well-written report. However, there are some relevant critical aspects I would like to address:

Major:

1. Methods, design and procedure: The parent study includes biomedical markers, interview data as well as subject's self-reports. Please present more details. Did the patients undergo a general medical examination at baseline? At one single or at different study centers? How many clinicians conducted the M.I.N.I.? Which profession? Did the clinicians conduct the M.I.N.I. over the phone or in person? After/before the medical examination? Did the subjects fill out the self-reporting questionnaires at the examination, or did they receive and return the questionnaires by mail? When?

2. Methods, design and procedure: There are some misleading inconsistencies in the description of the study design. The author states that this is a cross-sectional study (abstract line 25; discussion line 273), that this study used "data collected at baseline" (line 93) while, in fact, he also conducted a longitudinal analysis. This should be presented more consistently. Similarly, the author states that he also studied "a separate sample" (abstract, line 26), which might suggest that two completely different samples were used for this study, while this statement refers to the longitudinal analysis. This should be stated more clearly, too.

3. Methods, line 103: Follow-up data were collected at "first or second follow-up". Why these two occasions? How many subjects at first, how many at second follow-up? What is the time-line for first and second follow-up?

4. Methods, assessment, FSS: While current or past general medical conditions were assessed using a list of 30 medical illnesses, no such statement is given for the assessment of FSS. Did the investigators present a list of FSS, or did they assess just these four FSS? Furthermore, subjects were asked for current or past general medical conditions – what about FSS, current or past? Finally, if subjects were allowed to also refer to past general medical conditions (and past FSS?) – how can you ascertain that the analysis refers to current morbidity and comorbidity, as this is implied by the manuscript, from my point of view.

5. Methods, assessment, general health/role physical: This section was a little bit unclear to me. Did the author apply just a selected number of items from the SF-36? Typically, the subscales of the SF-36 are standardized on a scale ranging form 0 to 100.. Importantly, in the SF-36 some items are recalibrated. Here, the analysis seems to differ. If this study only included a subset of items from the SF-36, than the rationale should be given as this decision was taken giving up the comparability of the data for the SF-36. Moreover, the reliability and validity of this new scale is unclear. Please comment.

6. Methods, assessment: The assessment of neuroticism is not included, please add.

7. Methods, statistical analysis: What about missing data, and how did you deal with missings?

8. Results, line 178; table 1: The author states that data are shown for females only, "as there was a significantly higher rate of psychiatric disorder in females". This is quite unusual, as well as the justification for this procedure. From my point of view, this represents some kind of biased reporting. I would like to suggest presenting the complete data for the whole sample.

9. Results, univariate analysis: The author presents the results for IBD in table 3. Here, it is shown that migraine, IBS, and fibromyalgia were included as variables. From my point of view, it would be consequent to include all other five conditions, i.e. in this case also CFS, RA, and diabetes. By the way, why is migraine included here?

10. This leads me to the following remark: The author states that general medical conditions were assessed out of the list of 30 diseases. I guess that migraine was included in this list, but the rationale for selecting juft one further medical condition (migraine: FSS or not?) is unclear. Maybe it would be more useful to include the comorbidity score (sum of comorbid conditions, excluding the three conditions that are at the core of this analysis).

11. Results, multivariable analysis: The author applied backward elemination of variables. For me, it is unclear whether table 4 includes all variables entered in the multivarialbe logistic regression, or whether the variables presented are the results of backward elimination. Please clarify and state the full initial variable list for multivariable logistic regression.

12. Results, multivariable analysis, line 228: The author states that prior psychiatric disorder was the strongest predictor in all conditions. This statement is misleading, as the OR cannot be compared between the variables. The predictor variables included differ with regard to their metrics. As OR refers to the unit of the metric of each variable, and as the metrics differ, one cannot develop a hierarchy of influence on the outcome depending on the OR of the predictors.

Minor:

13. Introduction, lines 67-71: From my point of view, the statements in this para should be supported by some authoritative references, please refer to relevant work.

14. Methods, design and procedure: Please add the possible age range of the subjects in the Lifeline cohort.

15. Methods, assessemennt: Please add information on the response categories and the reliability of the measures in the current sample.

16. Results, table 1: Abbreviations should be explained (Marr; Work f/t, IBS,...). Further, the variables "chronic illness difficulties" and "social appreciation score" are unclear – how are these variables assessed? RAND items – expression unclar.

17. Results, table 1: The prevalence rates of dysthymia and MDD are unusually low – please make sure that these results are correct, and if so please discuss these results explicitly.

18. Discussion: It is a little bit unusual to include a section on strengths and limitations at the start of the discussion – the author might want to consider including this section at the end of the manuscript.

19. Discussion, line 277: "such a study is unlikely in the near future" – why?

20. Discussion, comparison with previous literature: This section largely lists the results of other studies, without referring to the own study and without interpreting similarites and differences. As such, parts of this section could be moved to the introductory section, in order to underline the research gap and to justify the current analysis.

Reviewer #2: 1. Abstract:

• Background is to short and it is good if definition of the outcome variable is included

• Line 25-26, “In a cross-sectional design logistic regression identified the variables most closely associated… “ better to rewrite it. Why cross-sectional design? Did you use it?

• Line 41, “Abbreviations” should be part of the declaration section of the manuscript.

2. Introduction:

• The magnitude of the problem should be described based on previously published literatures.

• You had better to include the gaps to which you want to fill and the aim of study in the introduction section of the manuscript.

3. Methods and tools:

• Who are your source population? Is

• What was the significance level of p value? It should be mentioned in the methods section.

• How valid and reliable your tools are?

4. Result:

• The continuous variables are presented as mean ± standard deviation (SD) in the tables. Were your data normally distributed? Have you checked the skewness? How much was it?

5. Discussion:

• Line 260-63, the statement “The variables closely associated with psychiatric disorder included those indicating predisposition to develop such disorders (neuroticism, and prior psychiatric disorder) as well as environmental variables including stressful life events and difficulties and chronic personal health difficulty.” Is incomplete and better to rewrite it.

• Line 267, avoid the statement “The strengths and limitations of this study must be recognized.”

• Strengths and limitations should be placed at the end of your discussion and just before conclusion.

• Poor justification of you findings. You have to justify more about your finding than justifying the others

6. Conclusion:

• What are your recommendations based on your current findings?

7. References

• Reference numbers 35, 36, 38, and 39 are relatively outdated and hence better to replace them with recent one if there is any.

8. You need to include your plan what did you do for those with severe problems if in case you find in your ethical approval? If you got such a study participant, it is unethical to disclose and leave them as they were.

9. You have to consider a major editorial issues of the whole document

10. The writing in English is very poor. There is poor use of punctuations and grammar in most parts of the text, I believe that an English language editor needs to revise the entire text before publication.

6. PLOS authors have the option to publish the peer review history of their article (what does this mean?). If published, this will include your full peer review and any attached files.

Reviewer #1: **Yes: **Andreas Dinkel

Reviewer #2: No

---

## [Author Response · Author response to Decision Letter 0]

2 Mar 2023

Response to reviewers - Psychiatric disorders comorbid with general medical illnesses and functional somatic disorders: The Lifelines cohort study 

Please see my response to the reviewers comments below

5. Review Comments to the Author

Reviewer #1: This study investigates determinants of psychiatric disorders in patients with functional vs. general medical illnesses. The author uses data from the Lifelines cohort, a large population-based study. The sample includes the impressive number of 122.366 adults. The results show that the variables associated with psychiatric morbidity were quite similar for patients with functional and general medical illness. A longitudinal corollary analysis showed that the prevalence of psychiatric disorder prior to the development of the functional and general medical conditions were quite similar to that of the participants with already established conditions. The author concludes that the increased rate of psychiatric disorder in patients with functional somatic syndromes seems to be evident before onset of the syndrome, which might be due to genetic as well as environmental factors.

This is is an interesting study with some unique aspects. It is a very large population-based sample. The study assessed functional syndromes as well as general medical conditions. Psychiatric diagnoses were established using a structured psychiatric interview with the majority of the participants. The main limitation refers to the self-report of FSS and general medical conditions. Furthermore, only a subset of mental disorders were assessed during the psychiatric interview.

Despite these limitations, this is a very important study, as it provides direct comparison between people with medically unexplained and medically explained conditions from the general population. This allows a more unbiased evaluation of possible determinants of psychiatric comorbidity in these two groups of disorders. The DSM-5 has abandoned the distinction between medically explained and medically unexplained symptoms. However, the phenomenen of medically unexplained symptoms won't vanish only because of the new diagnostic category in the DSM-5. Patients and clinicians will continue to differentiate between these two phenomenon in clinical practice. Thus, results on differences and similarities in these two groups are of high interest.

This is a well-written report. However, there are some relevant critical aspects I would like to address:

Major:

1. Methods, design and procedure: The parent study includes biomedical markers, interview data as well as subject's self-reports. Please present more details. Did the patients undergo a general medical examination at baseline? At one single or at different study centers? How many clinicians conducted the M.I.N.I.? Which profession? Did the clinicians conduct the M.I.N.I. over the phone or in person? After/before the medical examination? Did the subjects fill out the self-reporting questionnaires at the examination, or did they receive and return the questionnaires by mail? When? These are the responses from the programme centre in Groningen:

Did the patients undergo a general medical examination at baseline? It depends what you define a medical examination. They underwent for example anthropometry. (but not a full exam).

At one single or at different study centers? Several study centers were open at the same time. As a result, participants had the possibility to have their first visit (measurements) at one location and their second visit (urine/blood samples) at another location. But of course, most of the time participants went to the same location.

How many clinicians conducted the M.I.N.I.? It is difficult to give a number as we have had several rounds. But all doctor’s assistants were trained in the use of the MINI and, in the first round (data used in this study), they would all have belonged to the first cohort of trained medical professionals.

 Which profession? Doktersassistenten (I believe the English translation is doctor’s assistant but it might also be medical assistant)

Did the clinicians conduct the M.I.N.I. over the phone or in person? In person during first assessment and second assessment

After/before the medical examination? The MINI was conducted at the end of the visit. So after the medical examination

Did the subjects fill out the self-reporting questionnaires at the examination, or did they receive and return the questionnaires by mail? At the first assessment Lifelines worked with paper questionnaires. Participants were asked to hand in their first questionnaire during the first visit (data used for the cross-sectional part of the current study). For the second assessment, online questionnaires became more popular. The invitation to complete the questionnaire was send out before the visit and after the visit participants received a reminder. As a result, during the second assessment it was possible to have the self-reporting questionnaire before or after the MINI/visit.

An abbreviated version of these details has been added to the text on page 5 (lines 121-3).

2. Methods, design and procedure: There are some misleading inconsistencies in the description of the study design. The author states that this is a cross-sectional study (abstract line 25; discussion line 273), that this study used "data collected at baseline" (line 93) while, in fact, he also conducted a longitudinal analysis. This should be presented more consistently. Similarly, the author states that he also studied "a separate sample" (abstract, line 26), which might suggest that two completely different samples were used for this study, while this statement refers to the longitudinal analysis. This should be stated more clearly, too. 

I agree this was unclear. The abstract now reads (page 1 lines 25-30) : 

In a cross-sectional design logistic regression identified at baseline the variables most closely associated with current psychiatric disorder in participants with a pre-existing medical or functional condition. In a separate analysis the prevalence of psychiatric disorder prior to onset of these conditions was assessed. This was a longitudinal study with psychiatric disorder assessed at baseline in participants who subsequently developed a medical or functional condition between baseline and follow-up.

The text in the Methods section now reads (page 5 lines 130-135) :

The first set of analyses formed a cross-sectional study using baseline data only. The correlates of psychiatric disorder were determined among participants with pre-existing medical or functional conditions. The second, longitudinal study examined psychiatric disorder that was present at baseline before the onset of one of the medical or functional conditions; the 6 diagnostic groups were defined on the basis of medical or functional conditions which developed between baseline and first or second follow-up assessment.

3. Methods, line 103: Follow-up data were collected at "first or second follow-up". Why these two occasions? How many subjects at first, how many at second follow-up? What is the time-line for first and second follow-up? 

The following text has been inserted (page 6, lines 146-9.) 

At follow-up participants were asked “Since the last assessment have you developed…(list of medical and functional conditions)”. Participants were classified as a new onset of one of the 6 conditions if they had not recorded it at baseline and they recorded it at either follow-up. 

There is no theoretical reason to consider first and second follow-up assessments separately. The first follow-up occurred at 17 months after baseline and the second follow-up 29 months. Data from the two follow-ups were combined to capture all new onsets for the longitudinal analysis. 

4. Methods, assessment, FSS: While current or past general medical conditions were assessed using a list of 30 medical illnesses, no such statement is given for the assessment of FSS. Did the investigators present a list of FSS, or did they assess just these four FSS? 

The Lifelines cohort were asked only about the 3 functional somatic syndromes: irritable bowel syndrome, fibromyalgia and chronic fatigue syndrome. 

Furthermore, subjects were asked for current or past general medical conditions – what about FSS, current or past?

The questions were identical for FSS and general medical conditions. At baseline participants were asked about current and past disorders (medical and FSS); the questionnaire did not distinguish past and current conditions. At follow-up assessments participants were asked the same questions regarding the 3 FSS as part of the long list of general medical disorders.

 Finally, if subjects were allowed to also refer to past general medical conditions (and past FSS?) – how can you ascertain that the analysis refers to current morbidity and comorbidity, as this is implied by the manuscript, from my point of view. 

The study could not differentiate between current and past disorders but it was assumed that once a participant had developed inflammatory bowel disease, rheumatoid arthritis or diabetes they will continue to be affected. These are not diseases that disappear completely in the majority of people. Similarly for FSS. One could argue that the likelihood of psychiatric disorder varies whether the disease is in an active or quiescent phase but the study could not assess this. However, in a large population-based study of this nature one assumes that minor fluctuations are similar across all groups and there is no systematic bias. 

5. Methods, assessment, general health/role physical: This section was a little bit unclear to me. Did the author apply just a selected number of items from the SF-36? Typically, the subscales of the SF-36 are standardized on a scale ranging form 0 to 100.. Importantly, in the SF-36 some items are recalibrated. Here, the analysis seems to differ. If this study only included a subset of items from the SF-36, than the rationale should be given as this decision was taken giving up the comparability of the data for the SF-36. Moreover, the reliability and validity of this new scale is unclear. Please comment.

The Lifelines study questionnaire included the full SF-36 and all scores were adjusted according to the recommended scoring system. The results are often quoted as two overall summary scores: Mental and Physical component scores, which range from 0-100. However, these two summary scores are comprised of 8 scales , which provide a comprehensive profile of health status; four of these are related to physical aspects of impaired health status, which were the focus of the current study. There are data showing good internal reliability for these 8 scales; indeed, some think the psychometric properties of the 8 basic scales are preferable to the PCS and MCS summary scales (Jenkinson, Hann, Wilson,). 

Jenkinson C, Stewart-Brown S, Petersen S, Paice C. Assessment of the SF-36 version 2 in the United Kingdom. J Epidemiol Community Health. 1999 Jan;53(1):46-50.

Hann M, Reeves D.The SF-36 scales are not accurately summarised by independent physical and mental component scores. Qual Life Res. 2008 Apr;17(3):413-23. 

Wilson D, Parsons J, Tucker G. The SF-36 summary scales: problems and solutions. Soz Praventivmed. 2000;45(6):239-46..

The current study aimed to test the hypothesis that in general medical disorders (e.g. rheumatoid arthritis) the strongest correlations with psychiatric disorder would be with joint pains and impairment whereas in the functional somatic syndromes (e.g. IBS) psychiatric disorder would show strongest correlation with past psychiatric history and predisposing factors such as neuroticism, poor social support and stressful life events. 

In order to test this hypothesis the current study required measures of physical impairment and pain that arise from general medical disorders and functional syndromes symptoms but which are not closely associated with mental state. It was not, therefore appropriate to use the overall summary scores (0-100). Instead the physical function and role physical scales of the SF36 was used. Theses scales use data from 14 items to assess mobility, vigorous activities and personal care. The other two scales (bodily pain and general health) assess the degree to which bodily pain causes impairment and the individual’s perception of their health (e.g.”I seem to get sick a little easier than other people”). These scales measured exactly the dimensions of pain and impairment mentioned in the hypothesis. 

6. Methods, assessment: The assessment of neuroticism is not included, please add. 

This omission has been corrected (page 7, line 183) as has the omission of the social production function reference (page 8 line 187). 

7. Methods, statistical analysis: What about missing data, and how did you deal with missings?

My statistical advice was that with very large samples it is not necessary to deal with missing data. Please note that a companion paper, using the same dataset, found that of 80,888 participants followed up 65,904 (81.5%) had complete data for multiple regression analyses; missing values are therefore, unlikely to have affected the results. 

See: Creed F. The predictors of somatic symptoms in a population sample: The Lifelines cohort study Psychosom Med . 2022 Nov-Dec 01;84(9):1056-1066.

8. Results, line 178; table 1: The author states that data are shown for females only, "as there was a significantly higher rate of psychiatric disorder in females". This is quite unusual, as well as the justification for this procedure. From my point of view, this represents some kind of biased reporting. I would like to suggest presenting the complete data for the whole sample.

I think this refers to Figure 1 which is now mentioned on line 225. I pondered over this problem for some time. There was no intention to bias the reporting. I wanted to present a simple figure which allowed the reader to see which were the most common psychiatric diagnoses and whether psychiatric disorder was more frequent in the functional somatic syndromes compared to the general medical disorders. If I presented data for both sexes together this might bias the reporting as there was a higher prevalence of psychiatric disorder in women and the proportion of women was higher in some conditions than others (fibromyalgia = 91.5% female, RA =63.5% female). At least, in fig 1 as currently presented, the reader can see easily that the prevalence of GAD and panic disorder is slightly higher in fibromyalgia than RA at least in women. 

I have added another supplementary table (S1) to show the prevalence of psychiatric disorder in men and women separately for the 6 conditions. 

I think it should be for the editor to decide whether figure 1 remains as it stands or the data be presented in another way. 

In the meantime I have added the following data to the text to clarify why figure 1 only includes females. (page 9 lines 227-8)..

data are shown for females only as there was a significantly higher rate of psychiatric disorder in females and the proportion of females varied in each diagnostic group (fibromyalgia 91.5% females, IBS 81%, CFS 70.3%, IBD 64.5%, RA 63.5%, Diabetes 53.7%).

9. Results, univariate analysis: The author presents the results for IBD in table 3. Here, it is shown that migraine, IBS, and fibromyalgia were included as variables. From my point of view, it would be consequent to include all other five conditions, i.e. in this case also CFS, RA, and diabetes. By the way, why is migraine included here?

The 3 functional somatic syndromes (IBS, CFS and fibromyalgia) were included because of their known comorbidity with general medical disorders (Petersen 2018) so the association between IBD (for example) and psychiatric disorders could be mediated by the presence of an FSS. Thus FSS should be included as a possible covariate when examining the association between a medical or functional disorder and psychiatric disorder. IBD, RA and diabetes do not show such a close association with psychiatric disorder and are unlikely to act as important covariates in the association between FSS and psychiatric disorder. 

Petersen MW, Skovenborg EL, Rask CU, et al Physical comorbidity in patients with multiple functional somatic syndromes. A register-based case-control study. J Psychosom Res. 2018 Jan;104:22-28.

This is borne out in the univariable analyses. 

Tables 3, S3 and S6 show the results of univariate analyses prior to logistic regression analyses. These tables show a very clear association of IBS, CFS and fibromyalgia with psychiatric disorder in participants with IBD, RA and diabetes. Among participants with these general medical disorders the prevalence rates of IBS, CFS and fibromyalgia are often twice as high in those with psychiatric disorder compared to those without. Therefore it is appropriate to include IBS, CFS and fibromyalgia as independent variables in the logistic regression to see if they are correlates of psychiatric disorder in participants with general medical disorders. 

On the other hand Tables 3, S2-S6 show that “number of general medical conditions” (and the result is similar if IBD, RA and Diabetes are used as independent variables) is not clearly associated with psychiatric disorder in any of the 6 conditions. Although the differences between participants with and without psychiatric disorder appear to be significant this merely reflects the very large sample size; the actual differences between the two groups is minimal in contrast to that mentioned above for the FSS. Therefore IBS, CFS and fibromyalgia, but not IBD, RA and diabetes, were included in the logistic regression. Interestingly, this revealed that CFS was the only independent correlate of psychiatric disorder and only in participants with RA. 

I agree about migraine. I have removed migraine from tables 3 and S2-S6. It is often mentioned in the relevant literature as it is regarded as a pain condition but it is inappropriate to include it here.

10. This leads me to the following remark: The author states that general medical conditions were assessed out of the list of 30 diseases. I guess that migraine was included in this list, but the rationale for selecting juft one further medical condition (migraine: FSS or not?) is unclear. Maybe it would be more useful to include the comorbidity score (sum of comorbid conditions, excluding the three conditions that are at the core of this analysis).

As mentioned above, I have removed migraine for tables 3 and S1-S5. 

Unfortunately it is technically impossible for me to redo the analyses having removed the three general medical conditions from the comorbidity score. But the measure is intended to be a marker of all medical disorders on the basis that the more medical conditions an individual suffers the greater the psychological burden experienced. The greater the burden the more likely it is likely to be associated with psychiatric disorder. If a person with RA also has IBD and diabetes this should be represented in the data, in my view. I am afraid I do not understand the logic for removing certain medical comorbidities from the comorbidity score. 

11. Results, multivariable analysis: The author applied backward elemination of variables. For me, it is unclear whether table 4 includes all variables entered in the multivarialbe logistic regression, or whether the variables presented are the results of backward elimination. Please clarify and state the full initial variable list for multivariable logistic regression.

I apologise for this omission. The following text has now been added on page 8, lines 205-208):

The dependent variable was psychiatric disorder, present or absent. The independent variables were the all the variables listed in table 1 except the 5 DSM-IV and 3 general medical diagnoses (inflammatory bowel disease, rheumatoid arthritis and diabetes, which were included in the number of general medical illnesses score).

12. Results, multivariable analysis, line 228: The author states that prior psychiatric disorder was the strongest predictor in all conditions. This statement is misleading, as the OR cannot be compared between the variables. The predictor variables included differ with regard to their metrics. As OR refers to the unit of the metric of each variable, and as the metrics differ, one cannot develop a hierarchy of influence on the outcome depending on the OR of the predictors.

This is true and I fully accept the point. The word “strongest” has been removed 

Minor:

13. Introduction, lines 67-71: From my point of view, the statements in this para should be supported by some authoritative references, please refer to relevant work.

This has been done (see Reviewer 2, Point 2 – Introduction)

14. Methods, design and procedure: Please add the possible age range of the subjects in the Lifeline cohort.

The cohort includes children from 6 months of age up to 93 years of age but only those of 18 or over were included in this study. The upper age limit has been inserted in the text: Page 9 line 216.

15. Methods, assessemennt: Please add information on the response categories and the reliability of the measures in the current sample. 

There are many measures used in this study. They are described on pages 6 and 7 and each is referenced. The reader is also directed to the overall description of the Lifelines cohort study (refs 41,42). I have provided more detail of the scoring of two scales in the next point. 

. If there is a particular variable that the reviewer and editor consider needs further clarification perhaps this could be specified? 

16. Results, table 1: Abbreviations should be explained (Marr; Work f/t, IBS,...). Further, the variables "chronic illness difficulties" and "social appreciation score" are unclear – how are these variables assessed? RAND items – expression unclar.

I apologise for the abbreviations. These have now been corrected to read as follows page 10, line 232:

Few years of education (secondary or less)

Married or cohabiting

In full time employment 

Unable to work through illness 

Low income (lowest quintile)

The scoring of chronic illness difficulty is now explained on page 7 lines 169-171 as follows: The respondent was asked if their experience of their health was stressful (e.g. regularly ill, chronically ill) and they responded on a three point scale; (0=not stressful, 1=slightly stressful, 2=very stressful and a mean score quoted (48). The mean score was used in the analysis.

Neuroticism and social appreciation instruments are now explained more fully in the text (page 7 183- 189):

Neuroticism This was assessed using the Revised NEO Personality Inventory (NEO PI-R). The Lifelines questionnaire included the facets of anger/hostility, self-consciousness, impulsivity, and vulnerability. Each facet is assessed with eight items, scored on a five-point Likert scale that ranges from strongly disagree to strongly agree. (51).

Social Appreciation was assessed using the Social production function measure (SPF-IL). This is a 15 item scale asking about 6 dimensions of social support ; each item is scored on a 4 point scale and a high score represents good social support (52).

The description of the RAND instrument has been clarified (page 7 line 175-180): 

RAND instrument Current health status was assessed using the RAND 36-Item Health Survey General Health scale (49). Four of the 8 scales were used as they were relevant to this study: Physical Function (10 questions) and Role Physical (4 Qs), measures the degree to which daily life is restricted by physical limitations. Bodily pain (2Qs) assesses impairment due to pain. General Health (4 Qs) assesses how ill the individual feels. Each question is scored on a 3, 4 or 5-point Likert scale. On these scales a low or negative score indicates greater impairment .

17. Results, table 1: The prevalence rates of dysthymia and MDD are unusually low – please make sure that these results are correct, and if so please discuss these results explicitly.

I believe these results are correct as they concur with other reports from the Lifelines cohort ( Meurs, Hagen and Janssen ) Most population-based studies of affective disorders quote prevalence rates for the last 12 months whereas the MINI asks only about current disorders (last 2 weeks). This probably explains the lower rates in the present study. 

Meurs M, Roest AM, Wolffenbuttel BH, Stolk RP, de Jonge P, Rosmalen JG. Association of Depressive and Anxiety Disorders With Diagnosed Versus Undiagnosed Diabetes: An Epidemiological Study of 90,686 Participants. Psychosom Med. 2016 Feb-Mar;78(2):233-41. 

Hagen JM, Sutterland AL, da Fonseca Pereira de Sousa PAL, et al. Association between skin autofluorescence of advanced glycation end products and affective disorders in the lifelines cohort study. J Affect Disord. 2020 Oct 1;275:230-237. doi: 10.1016/j.jad.2020.06.040. Epub 2020 Jul 14.

Janssens KA, Zijlema WL, Joustra ML, Rosmalen JG. Mood and Anxiety Disorders in Chronic Fatigue Syndrome, Fibromyalgia, and Irritable Bowel Syndrome: Results From the LifeLines Cohort Study. Psychosom Med. 2015 May;77(4):449-57

18. Discussion: It is a little bit unusual to include a section on strengths and limitations at the start of the discussion – the author might want to consider including this section at the end of the manuscript.

This has been done

19. Discussion, line 277: "such a study is unlikely in the near future" – why?

There are very few studies worldwide that have assessed psychiatric disorder by research interview and include general medical and functional somatic syndromes. The closest is the UK biobank which includes studies of IBS and CFS but the measurement of mental disorder is by questionnaire only. See:

 Davis KAS, Coleman JRI, Adams M, et al Mental health in UK Biobank - development, implementation and results from an online questionnaire completed by 157 366 participants: a reanalysis. BJPsych Open. 2020 Feb 6;6(2):e18. doi: 10.1192/bjo.2019.100.

Lacerda EM, Mudie K, Kingdon CC, Butterworth JD, O'Boyle S, Nacul L. The UK ME/CFS Biobank: A Disease-Specific Biobank for Advancing Clinical Research Into Myalgic Encephalomyelitis/Chronic Fatigue Syndrome. Front Neurol. 2018 Dec 4;9:1026. doi: 10.3389/fneur.2018.01026. 

Wang K, Liu H, Liu J, et al Factors related to irritable bowel syndrome and differences among subtypes: A cross-sectional study in the UK Biobank. Front Pharmacol. 2022 Aug 26;13:905564. doi: 10.3389/fphar.2022.905564.

20. Discussion, comparison with previous literature: This section largely lists the results of other studies, without referring to the own study and without interpreting similarites and differences. As such, parts of this section could be moved to the introductory section, in order to underline the research gap and to justify the current analysis.

This has been done (see Reviewer 2, point 2 regarding introduction). 

The discussion has been revised 

Reviewer #2: 1. Abstract:

• Background is too short and it is good if definition of the outcome variable is included

Response: see below under “Introduction”

• Line 25-26, “In a cross-sectional design logistic regression identified the variables most closely associated… “ better to rewrite it. Why cross-sectional design? Did you use it?

On page 5 lines 130-135 the following text has clarified the cross-sectional and longitudinal components of the study:

The first set of analyses formed a cross-sectional study using baseline data only. The correlates of psychiatric disorder were determined among participants with pre-existing medical or functional conditions. The second, longitudinal study examined psychiatric disorder that was present at baseline before the onset of one of the medical or functional conditions; the 6 diagnostic groups were defined on the basis of medical or functional conditions which developed between baseline and first or second follow-up assessment.

• Line 41, “Abbreviations” should be part of the declaration section of the manuscript.

This has been done. 

2. Introduction:

• The magnitude of the problem should be described based on previously published literatures.

• You had better to include the gaps to which you want to fill and the aim of study in the introduction section of the manuscript.

The introduction has been considerably enlarged. The following text has been moved from the discussion and modified to broaden the background, explain the magnitude of the problem and describe the gaps in our knowledge which this study was design to fill:

On pages 3 (line 64 et seq) and 4 the following text has been added: 

The relationship between psychiatric disorders and general medical disorders may be more similar to that between psychiatric disorders and functional somatic syndromes than previously considered. Clarifying this relationship requires more refined methodology than that used in previous studies.

 In the existing literature the rate of psychiatric disorders among the physically ill tends to be overestimated because the studies have included clinical rather than population-based samples and have used self-administered questionnaires rather than research interviews to assess psychiatric disorders (2, 21, 23, 30, 31) . This prevents an accurate assessment of the relationship between psychiatric disorder and medical disorders (23, 32). Furthermore, the effect of psychiatric disorder on treatment-seeking and on symptom reporting on questionnaires may be greater in the functional somatic symptoms than general medical disorders (7, 33) . The current study aimed to assess, in a general population-based sample, the prevalence of psychiatric disorders in 3 functional somatic syndromes and 3 general medical disorders using a research psychiatric interview. It also aimed to assess whether the correlates of psychiatric disorder were similar in functional somatic syndromes and general medical disorders. To my knowledge no previous study has performed such a large population study using research interviews to assess psychiatric disorder. 

Prospective studies have shown that the prevalence of depression and other psychiatric disorders is higher than healthy controls both before and after the onset of inflammatory bowel disease, RA or diabetes and functional disorders (13, 17-27, 29, 30, 34-40) . The current study included a longitudinal analysis which assessed whether the prevalence of psychiatric disorder was similar before and after the onset of inflammatory bowel disease, RA, diabetes, IBS, CFS and fibromyalgia .

 The first analyses in this study examined whether the presence of psychiatric disorder was closely associated a) with symptoms and impairment, which result from medical or functional conditions, and/or b) with predisposition to develop psychiatric disorder, which would be evident before the onset of the medical or functional disorder. Thus it might be hypothesised that in general medical disorders (e.g. rheumatoid arthritis) the strongest correlations with psychiatric disorder would be with joint pains and consequent impairment whereas in the functional somatic syndromes (e.g. IBS) psychiatric disorder would show strongest correlation with past psychiatric history and predisposing factors for psychiatric disorder such as neuroticism, poor social support and stressful life events.

3. Methods and tools:

• Who are your source population? Is The general population. See page 5 (lines 111-117 )

The data used in this study came from the Lifelines study, a multi-disciplinary prospective population-based cohort study examining in a unique three-generation design the health and health-related behaviours of 167,729 individuals living in the north of the Netherlands . The study assessed at baseline a broad range of biomedical, socio-demographic, behavioural, physical and psychological variables which may contribute to future health outcomes (41, 42). People with low educational attainment and smokers are somewhat under-represented in the Lifelines cohort but otherwise it is representative of the total Dutch population (43).

• What was the significance level of p value? It should be mentioned in the methods section.

This omission has been corrected. The following text has been added to the statistical method section on page 8 (line 201) 

 “The value of p was set at 0.001 in view of the large number of variables that were assesses as correlates of psychiatric disorder.

• How valid and reliable your tools are?

References 35 - 39 and 49-52 provide data concerning the validity and reliability of the instruments used in this study.

4. Result:

• The continuous variables are presented as mean ± standard deviation (SD) in the tables. Were your data normally distributed? Have you checked the skewness? How much was it?

My statistical advice suggested that skewness is not an issue with very large numbers , in which parametrical statistical tests can be used regardless of normality.

5. Discussion:

• Line 260-63, the statement “The variables closely associated with psychiatric disorder included those indicating predisposition to develop such disorders (neuroticism, and prior psychiatric disorder) as well as environmental variables including stressful life events and difficulties and chronic personal health difficulty.” Is incomplete and better to rewrite it.

The first paragraph of the discussion (line 310 et seq) has been rewritten and this sentence has been removed

• Line 267, avoid the statement “The strengths and limitations of this study must be recognized.”

This has been done

• Strengths and limitations should be placed at the end of your discussion and just before conclusion.

This has been done

• Poor justification of you findings. You have to justify more about your finding than justifying the others

The whole discussion has been reworked so the previous work is now described in the introduction. Only pertinent points from the previous literature are now included in the discussion. There is greater evaluation of the current results. See line 318 et seq 

6. Conclusion:

• What are your recommendations based on your current findings?

These have been spelt out in a new paragraph (page 23, line 478): 

There are two recommendations arising as a result of this study The first is that future work concerning the relationship between psychiatric disorder and functional somatic syndromes be considered alongside the same relationship with general medical disorders, in which depression may accompany the disorder without necessarily being a causal factor. This should help avoid premature conclusions that psychiatric disorder is a principal factor causing the functional somatic syndromes (61). Secondly, in both research and clinical work we should be aware of possible interactions between psychiatric disorders and other behavioural and metabolic variables to identify the true role of anxiety and depression in the causation of the functional somatic syndromes.

7. References

• Reference numbers 35, 36, 38, and 39 are relatively outdated and hence better to replace them with recent one if there is any.

These are references to the instruments used in the study. As requested by reviewer 1 these have been retained and the description of the scoring systems been spelt out more fully. 

8. You need to include your plan what did you do for those with severe problems if in case you find in your ethical approval? If you got such a study participant, it is unethical to disclose and leave them as they were.

The Lifelines excluded any individuals who had severe physical or mental problems (page 5, line 117-9.

9. You have to consider a major editorial issues of the whole document

I have done this.

10. The writing in English is very poor. There is poor use of punctuations and grammar in most parts of the text, I believe that an English language editor needs to revise the entire text before publication.

I am not sure how to answer this point. According to Pubmed I have published 280 articles in high quality scientific journals and my use of English has not been questioned previously. However, I accept the criticism that some of the sentences are long and complex. I have reviewed the whole manuscript to ensure that it is written in short, clear sentences so it can be easily understood by any reader. 

6. PLOS authors have the option to publish the peer review history of their article (what does this mean?). If published, this will include your full peer review and any attached files.

I have no objection to this if the editor wishes this. 

Do you want your identity to be public for this peer review? For information about this choice, including consent withdrawal, please see our Privacy Policy.

Reviewer #1: Yes: Andreas Dinkel

Reviewer #2: No

---

## [Decision Letter · Decision Letter 1]

13 Apr 2023

PONE-D-22-33955R1Psychiatric disorders comorbid with general medical illnesses and functional somatic disorders: The Lifelines cohort studyPLOS ONE

Dear Dr. Creed,

Thank you for submitting your manuscript to PLOS ONE. After careful consideration, we feel that it has merit but does not fully meet PLOS ONE’s publication criteria as it currently stands. Therefore, we invite you to submit a revised version of the manuscript that addresses the points raised during the review process.

.

We look forward to receiving your revised manuscript.

Kind regards,

Wudneh Simegn Belay, MSc

Academic Editor

PLOS ONE

Additional Editor Comments:

The authors should address all comments raised by the reviewers. I recommend the authors to provide point by point response and incorporate to the revised manuscript.

Reviewers' comments:

Reviewer's Responses to Questions

**Comments to the Author**

1. If the authors have adequately addressed your comments raised in a previous round of review and you feel that this manuscript is now acceptable for publication, you may indicate that here to bypass the “Comments to the Author” section, enter your conflict of interest statement in the “Confidential to Editor” section, and submit your "Accept" recommendation.

Reviewer #1: (No Response)

Reviewer #2: All comments have been addressed

2. Is the manuscript technically sound, and do the data support the conclusions?

Reviewer #1: Yes

Reviewer #2: Yes

3. Has the statistical analysis been performed appropriately and rigorously? 

Reviewer #1: Yes

Reviewer #2: Yes

4. Have the authors made all data underlying the findings in their manuscript fully available?

Reviewer #1: No

Reviewer #2: Yes

5. Is the manuscript presented in an intelligible fashion and written in standard English?

Reviewer #1: Yes

Reviewer #2: Yes

6. Review Comments to the Author

Reviewer #1: Thank you for your efforts in revising the manuscript and your detailed and thoughtful response to the critical aspects I raised.

Only a few aspects remain:

1. Introduction, p. 4, lines 87-92. Here, the author gives a justification for the hypothesis that different associations with psychiatric disorder exist in persons with mediccal conditions vs. those with functional disorders. I am sorry, but I did not understand the reasoning. Why the strongest correlation with joint pain in medical conditions and, among others, neuroticism in FSS? - This seems quite arbitrary. At least, some references supporting this argumentation seem to be necessary.

2. Introduction, p. 4, lines 95-99: Please support your statement on known predisposing and precipitating factors for psychiatric illness with authoritative references.

3. Methods, measures, p. 6, line 143-149/line 150-151: As far as I understood from the author's response, the list of 30 current of past medical illnesses also included the three functional disorders of interest. - If this is correct, this should be stated more clearly.

4. Methods, measures, p. 7, RAND instruments: The author responded to my comment on the standardization of the SF-36 items that all scores were computed and adjusted according to the recommended scoring system. However, table 1 stills shows values for four subscales of the SF-36 that do not adhere to the common metric (0-100). In fact, the mean of the subscale general health is negative ! - Thus, I cannot see that the standard procedure was applied to compute these four subscales of the SF-36. To my knowledge, the current scores cannot be compared to the multitude of studies that applied the SF-36, and the current scoring is not transparent. I'd like to suggest correcting the analyses using the SF-36 subscales.

5. Methods, measures: The author does not present data on the reliability of the measures. Information on Cronbach's alpha of all measures in the current sample would be helpful.

6. Methods, p. 8, statistical analysis, line 207: Typo - were the all the.

Reviewer #2: (No Response)

7. PLOS authors have the option to publish the peer review history of their article (what does this mean?). If published, this will include your full peer review and any attached files.

Reviewer #1: **Yes: **Andreas Dinkel

Reviewer #2: **Yes: **Mengistie Diress

---

## [Author Response · Author response to Decision Letter 1]

4 May 2023

Response to reviewers for PONE-D-22-33955R1 Psychiatric disorders comorbid with general medical illnesses and functional somatic disorders: The Lifelines cohort study

Thank you for the kind words regarding my previous revisions. Please find below the details of the further revisions I have made in the light of the reviewer’s additional points. 

1. Introduction, p. 4, lines 87-92. Here, the author gives a justification for the hypothesis that different associations with psychiatric disorder exist in persons with mediccal conditions vs. those with functional disorders. I am sorry, but I did not understand the reasoning. Why the strongest correlation with joint pain in medical conditions and, among others, neuroticism in FSS? - This seems quite arbitrary. At least, some references supporting this argumentation seem to be necessary.

A new paragraph has been added to the introduction order to clarify the underlying purpose of this study (pages 3-4 lines 82-97):

The first analyses in this study examined whether psychiatric disorder was closely associated with a) the symptoms and impairment which accompany a medical or functional disorder, and/or b) a predisposition to develop psychiatric disorder, including neuroticism, psychosocial stress and prior psychiatric disorder which would be evident before the onset of the medical or functional disorder. Thus, for example, in rheumatoid arthritis it has been reported that the dominant predictors of self-reported depression are pain and fatigue and it is considered that these may be the main cause of the depression (41, 42). Depression in fibromyalgia is also associated with severity of pain and degree of impairment but, in addition, depression in fibromyalgia is closely associated with psychological variables such as neuroticism, learned helplessness, low self-efficacy and with psychosocial problems(43-46). These psychological variables are associated also with depression in the absence of physical illness (47-49). No previous study has assessed, across a range of disorders, whether depression comorbid with physical illness is associated both with the symptoms and impairment of the physical illness and with the psychological variables of neuroticism, prior psychiatric disorder, low social support and stressful life events unrelated to physical illness. In this study it was hypothesised that the correlates of psychiatric disorders would be similar in participants with functional somatic syndromes and general medical illnesses.

2. Introduction, p. 4, lines 95-99: Please support your statement on known predisposing and precipitating factors for psychiatric illness with authoritative references. This has been done (page 6, line 92). References 47-50 inserted:

 47. Bonsaksen T, Grimholt TK, Skogstad L, Lerdal A, Ekeberg Ø, Heir T, et al. Self-diagnosed depression in the Norwegian general population - associations with neuroticism, extraversion, optimism, and general self-efficacy. BMC public health. 2018;18(1):1076.

48. Kendler KS, Gardner CO, Prescott CA. Toward a comprehensive developmental model for major depression in women. Am J Psychiatry. 2002;159(7):1133-45.

49. Uliaszek AA, Zinbarg RE, Mineka S, Craske MG, Sutton JM, Griffith JW, et al. The role of neuroticism and extraversion in the stress-anxiety and stress-depression relationships. Anxiety Stress Coping. 2010;23(4):363-81.

50. Kendler KS, Gardner CO, Prescott CA. Toward a comprehensive developmental model for major depression in men. Am J Psychiatry. 2006;163(1):115-24.

3. Methods, measures, p. 6, line 143-149/line 150-151: As far as I understood from the author's response, the list of 30 current of past medical illnesses also included the three functional disorders of interest. - If this is correct, this should be stated more clearly.

This has been made explicit (page 6, line 151-2):

“It included IBS, fibromyalgia and chronic fatigue syndrome.”

4. Methods, measures, p. 7, RAND instruments: The author responded to my comment on the standardization of the SF-36 items that all scores were computed and adjusted according to the recommended scoring system. However, table 1 stills shows values for four subscales of the SF-36 that do not adhere to the common metric (0-100). In fact, the mean of the subscale general health is negative ! - Thus, I cannot see that the standard procedure was applied to compute these four subscales of the SF-36. To my knowledge, the current scores cannot be compared to the multitude of studies that applied the SF-36, and the current scoring is not transparent. I'd like to suggest correcting the analyses using the SF-36 subscales.

Apologies. This has been done in tables 1, 3 and S2-S6. 

5. Methods, measures: The author does not present data on the reliability of the measures. Information on Cronbach's alpha of all measures in the current sample would be helpful.

This requires access to raw data which are mostly unavailable to me now. I have been able to add the results for RAND items only – see page 8, Lines 187-8: 

“Cronbach’s alpha for these items in this sample were: physical functioning = 0.87, Role functioning/ Physical = 0.89, Bodily Pain = 0.81, General Health = 0.80.”

6. Methods, p. 8, statistical analysis, line 207: Typo - were the all the. 

This has been corrected Page 9 line 214

---

## [Decision Letter · Decision Letter 2]

16 May 2023

Psychiatric disorders comorbid with general medical illnesses and functional somatic disorders: The Lifelines cohort study

PONE-D-22-33955R2

Dear Dr. Creed,

We’re pleased to inform you that your manuscript has been judged scientifically suitable for publication and will be formally accepted for publication once it meets all outstanding technical requirements.

Kind regards,

Wudneh Simegn Belay, MSc

Academic Editor

PLOS ONE

---

## [Editor Report · Acceptance letter]

19 May 2023

PONE-D-22-33955R2 

Psychiatric disorders comorbid with general medical illnesses and functional somatic disorders: The Lifelines cohort study 

Dear Dr. Creed:

I'm pleased to inform you that your manuscript has been deemed suitable for publication in PLOS ONE. Congratulations! Your manuscript is now with our production department. 

Kind regards, 

on behalf of

Dr. Wudneh Simegn Belay 

Academic Editor

PLOS ONE